# Mars, a molecule archive suite for reproducible analysis and reporting of single-molecule properties from bioimages

**Nadia M Huisjes[1†], Thomas M Retzer[1,2†], Matthias J Scherr[1], Rohit Agarwal[1,2], Lional Rajappa[1], Barbara Safaric[1], Anita Minnen[1], Karl E Duderstadt[1,2]\***

[1]Structure and Dynamics of Molecular Machines, Max Planck Institute of Biochemistry, Martinsried, Germany; [2]Physik Department, Technische Universität München, Garching, Germany

**\*For correspondence:**
duderstadt@biochem.mpg.de

[†]These authors contributed equally to this work

**Competing interest:** The authors declare that no competing interests exist.

**Abstract** The rapid development of new imaging approaches is generating larger and more complex datasets, revealing the time evolution of individual cells and biomolecules. Single-molecule techniques, in particular, provide access to rare intermediates in complex, multistage molecular pathways. However, few standards exist for processing these information-rich datasets, posing challenges for wider dissemination. Here, we present Mars, an open-source platform for storing and processing image-derived properties of biomolecules. Mars provides Fiji/ImageJ2 commands written in Java for common single-molecule analysis tasks using a Molecule Archive architecture that is easily adapted to complex, multistep analysis workflows. Three diverse workflows involving molecule tracking, multichannel fluorescence imaging, and force spectroscopy, demonstrate the range of analysis applications. A comprehensive graphical user interface written in JavaFX enhances biomolecule feature exploration by providing charting, tagging, region highlighting, scriptable dashboards, and interactive image views. The interoperability of ImageJ2 ensures Molecule Archives can easily be opened in multiple environments, including those written in Python using PyImageJ, for interactive scripting and visualization. Mars provides a flexible solution for reproducible analysis of image-derived properties, facilitating the discovery and quantitative classification of new biological phenomena with an open data format accessible to everyone.

## Editor's evaluation

This is a valuable paper that reports an open-source platform for the storage and processing of single-molecule, camera-based, imaging data. The development and testing of the platform are very compelling and the platform will facilitate data sharing and reproducibility and will be of great interest to practitioners of single-molecule imaging experiments, both experienced and new to the field. The work represents significant and important steps towards unifying and standardizing how the field stores and processes data and expanding the base of researchers who can easily employ single-molecule imaging methods.

## Introduction

Reproducible analysis of bioimaging data is a major challenge slowing scientific progress. New imaging techniques generate datasets with increasing complexity that must be efficiently analyzed, classified, and shared. This challenge has gained wide recognition (*Carpenter et al., 2012*; *Eliceiri*

*et al., 2012*; *Lerner et al., 2021*; *Meijering et al., 2016*; *Ouyang and Zimmer, 2017*), which has led to the development of software frameworks for curating images (*Allan et al., 2012*; *Kvilekval et al., 2010*) and analysis workflows (*Rubens et al., 2020*), metadata reporting standards (*Goldberg et al., 2005*) and the creation of public archives to increase data availability (*Williams et al., 2017*). However, few standards exist for the reproducible analysis and reuse of image-derived properties and those that have been developed for single molecule fluorescence (*Greenfeld et al., 2015*; *Ingargiola et al., 2016*) offer limited options for adaption to other experimental configurations. There is an increasingly powerful toolkit to quantitatively follow the time evolution of the position, shape, composition, and conformation of individual cells and complexes, but these precious information-rich observations are generated in heterogenous formats that do not provide easy, transparent access to key features. As a consequence, the promise of new technologies is often unrealized, and new biological phenomena remain undiscovered in existing datasets, due to the lack of tools that enable robust classification and interactive exploration.

Single-molecule techniques provide access to rare intermediates in complex, multistage molecular pathways. Multicolor fluorescence imaging has revealed the conformational dynamics of membrane transport (*Akyuz et al., 2013*; *Erkens et al., 2013*), molecular states underlying assembly and transcription by RNA polymerase (*Baek et al., 2021*; *Duchi et al., 2016*), and DNA replication dynamics (*Duderstadt et al., 2016*; *Scherr et al., 2018*; *Ticau et al., 2015*). These approaches have been combined with spatial tracking on biological structures to clarify how exchange events and conformational changes modulate the function of motor proteins, as well as replication, transcription, and DNA repair machineries (*Crickard et al., 2020*; *Lewis et al., 2020*; *Niekamp et al., 2021*; *Scherr et al., 2022*). Biological macromolecules are routinely attached to microspheres which allow for controlled studies of the forces and torques involved in basic biological reactions (*Agarwal and Duderstadt, 2020*; *Dulin et al., 2013*; *Neuman and Nagy, 2008*; *Revyakin et al., 2006*). And finally, improvements in the sensitivity of camera sensors, fluorophore brightness, and new illumination strategies *Gao et al., 2012*; *Tokunaga et al., 2008* have enabled time-resolved studies of biological processes in live cells. These developments have revealed frequent exchange of factors during normal operation of replisomes (*Beattie et al., 2017*; *Kapadia et al., 2020*) and the dynamics of the transcription-factor target site search (*Chen et al., 2014*).

The discovery of new biological phenomena from these multidimensional observations, depends on long, multistage image analysis workflows, followed by careful grouping and feature classification. Images are typically preprocessed, to correct for non-uniform beam profiles and filtered to enhance detection of individual molecules and structures over the background. Additional tools are then used to follow the properties of individual biomolecules through time. These often provide some

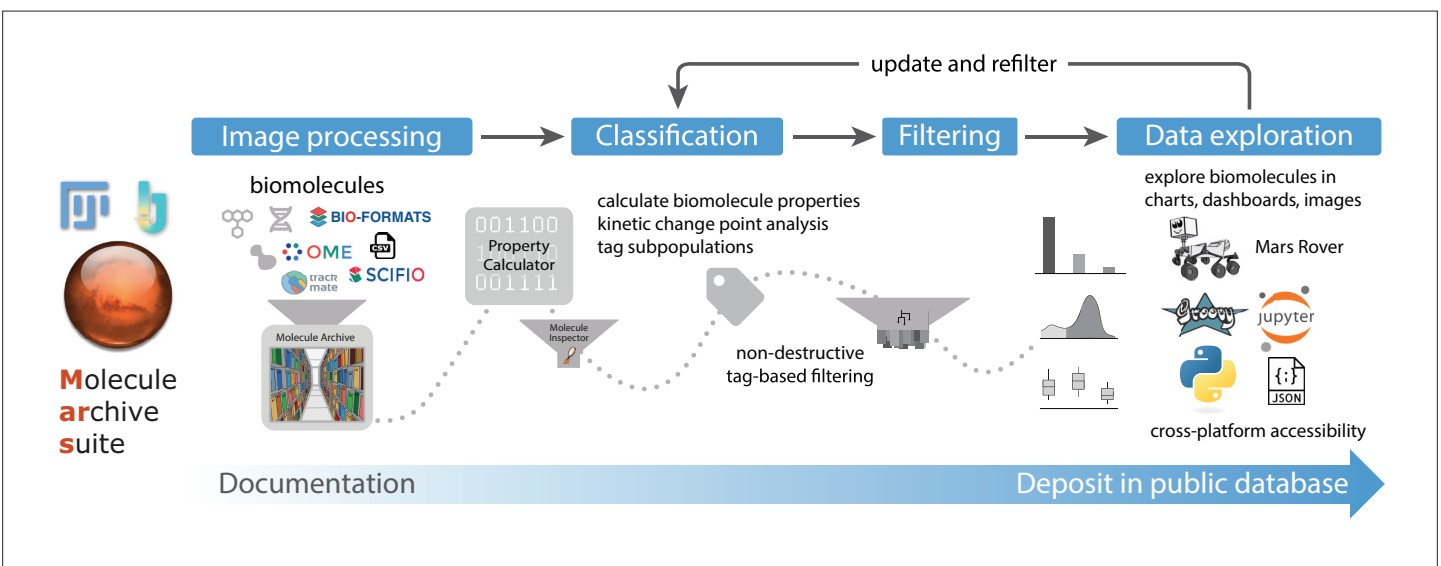

**Figure 1.** Overview of Mars workflows. The process of reproducible data analysis with Mars starting from image processing to iterative rounds of classification and filtering to the final stage of data exploration and deposition into a public database.

form of tabular data containing aggregated results, which are either manually evaluated or filtered using a feature unique to the desired group of observations. Typically, at this point, results can no longer be re-evaluated in the context of the original images due to unidirectional transformations or lack of interactive image evaluation software. The final data are frequently migrated to another platform ideally suited for the generation of publication-quality figures. This necessitates further data restructuring during transfers between software platforms. Large datasets are downsized to cope with limited storage by removal of rejected observations in ways that irreversibly alter datasets. As rejection criteria change and additional data must be incorporated, precious time and reproducibility are lost. This process is often repeated for each experimental application and research group, leading to substantial duplication of efforts.

Recognizing these issues in common single-molecule image processing workflows, we developed Mars, which provides a collection of Fiji/ImageJ2 (*Pietzsch et al., 2012*; *Rueden et al., 2017*; *Schindelin et al., 2012*) commands and integrations with well-established Fiji workflows for processing images and image-derived properties based on a versatile Molecule Archive architecture (*Figure 1*). This architecture allows for seamless virtual storage, merging, and multithreaded processing of very large datasets. A simple, yet powerful interface allows for fast and easy access to subsets of records based on any biomolecule property. This framework allows for the same data structure to be used from initial image processing all the way to the generation of final figures. All image metadata and biomolecule records are assigned universally unique ids (uids) upon creation, which together with comprehensive logging, ensures the history of each record remains traceable through long and complex analysis workflows involving numerous data merging steps. Non-destructive tag-based filtering ensures no observations are lost and updates to rejection criteria only require retagging of processed records. These design principles facilitate the reproduction of analysis workflows and ensure Molecule Archives provide a comprehensive, transparent format for the deposition of final datasets.

The modular design of commands and minimal definitions of image metadata and biomolecule records ensure flexibility that facilitates the development of varied Mars workflows. To illustrate the range of image processing and kinetic analysis tasks that can be accomplished, we present the results of three applications. We demonstrate the versatility of Mars record types in representing both single molecules as well as large macromolecular structures by tracking single RNA polymerases on long DNAs. We benchmark a multichannel fluorescence integration workflow using well-established single-molecule FRET frameworks. And finally, we show how high-throughput imaging of DNA-tethered microspheres reveals the results of complex topological transformations induced by controlled forces and torques. In each example, we highlight image processing commands and methods for accessing and manipulating Molecule Archives using scripts. Each workflow provides a basic framework that can be further adapted to custom applications by the introduction of additional processing steps and modification of the scripts provided. The graphical user interface provided by the Mars Rover facilitates workflow refinement through interactive data exploration and manual classification with multichannel image views. Taken together, these features make Mars a powerful platform for the development of imaging-processing workflows for the discovery and quantitative characterization of new biological phenomena in a format that is open to everyone.

## Results

### Common pitfalls in single-molecule image-processing workflows

Single-molecule imaging platforms provide an increasingly powerful toolkit to observe complex biological systems, but obtaining quantitative information depends on multistage analysis workflows with many potential pitfalls. The core design principles of Mars were developed to avoid common issues. Before introducing the architecture of Mars in depth, we will illustrate the benefit of Mars with a practical example. For those not yet experienced with single-molecule imaging, this example should help to clarify how Mars improves data readability and workflow reproducibility to avoid issues that frequently arise during dataset transformations when analysis steps are not properly documented and annotated.

Many common single-molecule experiments involve recording images of the position and intensity of fluorescent labels attached to biomolecules as a function of time. For example, consider the simple experiment of fluorescently labeled polymerases traveling along DNA molecules during synthesis

(covered in greater depth in workflow 1). Experimenters typically want to track polymerase position and fluorescence intensity in consecutive images. From the raw images, they would like to determine the polymerase synthesis rate, product sizes, and number of polymerases on each DNA molecule. This is accomplished by finding the locations of fluorescent spots in individual images, followed by linking spots from the same molecules over time. In a typical single-molecule workflow, tracking results are generated in a single table with columns for the image number, molecule position, and an index number for each track. Several filters are used to select moving polymerases that travel a minimum distance and have an intensity consistent with a single dye. After this automated filtering, tracks are manually checked to remove those with tracking errors. In a typical workflow, these filtering steps remove rejected tracks from the table. Finally, a conversion factor that relates DNA length to polymerase coordinates is used to calculate the synthesis rate and product length. The process is then repeated for many videos generating a final merged table with reindexed global track numbers.

This typical workflow has several pitfalls that increase the analysis time, make discovery of new phenomena more difficult, lead to irreproducibility, and, in the worst case, result in mistakes. The most significant problem is that every step is unidirectional. Tracks rejected during automated or manual filtering are removed. As a consequence, if any filtering criteria change, the analysis must be repeated from the start. Moreover, no framework is provided to easily document which scripts and thresholds were used and why tracks were manually rejected. The most significant pitfall arises from merging multiple videos and reindexing track numbers. This makes it impossible to evaluate the final dataset in the context of the original raw videos and their metadata. For example, when a mistake is discovered in a subset of experiments, such as poor polymerase labeling, incorrect temperature, illumination, or contamination, there is no way to easily remove the subset of experiments from the final dataset since the tracks have no identity linking them to the original videos. Beyond these immediate practical pitfalls, the structure of the workflow makes it difficult to ask complex questions that relate one biomolecule property to another and quickly re-evaluate the dataset based on a new model. For

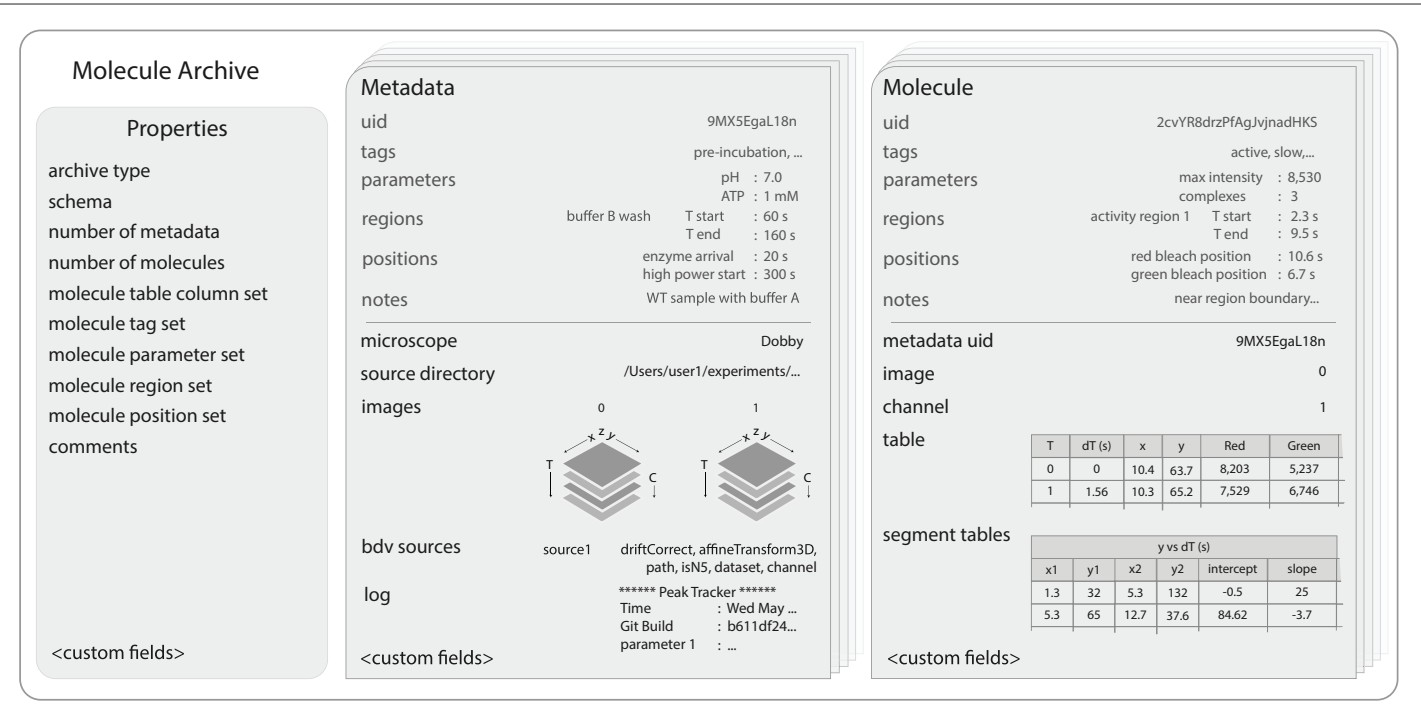

**Figure 2.** Molecule Archive structure. Schematic representation of the structure of Molecule Archives consisting of three types of records: *Properties*, *Metadata*, and *Molecule*. The single *Properties* record contains global information about the Molecule Archive contents, the *Metadata* records store information about the images used for biomolecule analysis (e.g. image dimensions, the analysis log), and the *Molecule* records store molecule-specific information (e.g. position over time, intensity).

The online version of this article includes the following figure supplement(s) for figure 2:

**Figure supplement 1.** Screenshots of the Mars rover window.

example, how does the synthesis rate depend on the number of polymerases? How does polymerase number influence the frequency of pauses? What if there are large differences in DNA extension between molecules or videos that require unique correction factors. The entire analysis and filtering would need to be done again.

Mars was developed to address these common pitfalls that arise in workflows like the example provided. First, Mars provides image processing commands streamlined for single-molecule applications that generate Molecule Archives containing a record of the analysis steps. Filtering is done using tags without the removal of rejected data to avoid unidirectional workflows. When filter criteria change only the tags need to be updated without redoing the analysis. Different subsets of existing tags may also be used to study complex differences between subpopulations such as polymerase clusters. The uids given to records upon creation, together with unique metadata ids, ensure a traceable history when datasets are merged, so that final datasets can always be evaluated in the context of the original raw images. The biomolecule storage and graphical interface in Mars are built to simplify the exploration of complex relationships based on different biomolecule properties. If experimental problems are discovered in a subset of videos, they can easily be filtered using the metadata ids stored with the records. All of these elements provide endless reslicing possibilities using the same dataset that overcome common pitfalls in single molecule image processing workflows.

## Molecule Archive architecture

Molecule Archives provide a flexible standard for storing and processing image-derived properties adaptable to a broad range of experimental configurations. Molecule Archives contain three record types: *Properties*, *Metadata*, and *Molecule* (*Figure 2*). Each type is defined in an interface, independent of implementation details. This abstraction ensures Molecule Archives support a variety of biomolecule, metadata, and property implementations. Moreover, this allows for the creation of new implementations that seamlessly work with the existing code base, algorithms, and user interface. To simplify the mechanics of record retrieval and the process of dataset merging, all *Molecule* and *Metadata* records are assigned human-readable, base58-encoded universally uids. Storing records using uids reduces indexing requirements, facilitates scalable processing using uid-to-record maps that support multithreaded operations, and ensures the traceability of records through analysis workflows.

Molecule Archives contain a single *Properties* record with the type of the Molecule Archive and global information about the Molecule Archive contents. This includes the number of *Metadata* and *Molecule* records and unique names used for tags, parameters, regions, positions, and table columns. The *Properties* record stores global Molecule Archive comments typically containing important information about the analysis strategy and naming scheme for tags, parameters, regions, positions, and other fields to orient a new researcher that did not perform the original analysis. To improve the organization and readability of comments, the Mars Rover also provides a convenient Markdown editor.

*Metadata* records contain experimental information about the images used for biomolecule analysis. To ensure compatibility with the broadest set of image formats and maximum reusability, image dimensionality, timepoints, channels, filters, camera settings, and other microscope information are stored in OME format (*Goldberg et al., 2005*). Molecule Archives do not store the raw images, only the image-derived properties and the source directory containing the project. Nevertheless, interactive image views linked to molecule records are supported through big data viewer (BDV) integration (*Pietzsch et al., 2015*) with HD5 and N5 (*Stephan Saalfeld et al., 2022*) formatted images. The image file locations, coordinate transforms, and other settings are stored in *Metadata* records. Global events in time influencing all biomolecules are documented using *Metadata* regions and positions that are available for kinetic analysis and scripting. Global experimental conditions are stored as parameters in the form of key-value pairs (e.g. buffer composition and temperature). *Metadata* records are categorized using tags to filter biomolecules by whole experiments. The *Metadata* records also contain a log where all commands and settings used for processing are recorded, thus maintaining the entire history of data processing throughout the analysis.

*Molecule* records contain fields for convenient storage of common image-derived properties of biomolecules. This includes a table that typically contains the position and intensity over time, the uid of the *Metadata* record containing the primary experimental information, and the index of the image containing the biomolecule. Events of interest in time can be marked with regions and positions (e.g. activity bursts, dye bleaching), which allow for event-specific calculations and kinetic analyses.

Calculated global biomolecule properties are stored as parameters in the form of key-value pairs (e.g. mean intensity, distance traveled, and mean position). Mars comes with kinetic change point (KCP) commands for unbiased identification of distinct linear regimes and steps (e.g. polymerase synthesis rate, FRET states, and dye bleaching) (*Hill et al., 2018*), which are stored in segment tables. And finally, Molecule records are categorized for later analysis using custom tags and notes for manual assessments.

Molecule Archives can be created, saved, and reloaded in a variety of formats in multiple environments both desktop-based and without a user interface for parallel processing on high-performance clusters. Molecule Archives are saved in JSON format using the field schema outlined in *Figure 2*. By default, Molecule Archives are written to single files with a yama extension using smile encoding and compression of molecule table data to reduce file size, but can also be saved and reopened in plaintext JSON. Mars supports processing of very large datasets, that typically do not fit in physical memory using a virtual storage mode in which records are retrieved only on-demand supported by a simple filesystem-backed record hierarchy. This architecture provides a powerful and flexible framework for multistep analysis workflows involving very large datasets.

## Mars Rover–interactive molecule feature exploration and image views

The discovery of new biological phenomena using single-molecule techniques often relies on manual exploration of individual biomolecules in primary images together with image-derived measurements. To simplify this process, we developed a user interface in JavaFX, called the Mars Rover, that provides access to all information stored in Molecule Archives. The Mars Rover is integrated into Fiji with windows available for all open Molecule Archives. Open Molecule Archives are available as inputs and outputs in Fiji/ImageJ2 commands and in supported scripting languages.

In the Mars Rover, Molecule Archive windows contain tabs and subpanels that provide complete access to all fields of *Molecule* and *Metadata* records as well as global properties, comments, and interface settings (*Figure 2—figure supplement 1*). A customizable global dashboard provides information about the Molecule Archive contents and scriptable chart widgets that can be adapted to specific workflows. The *Molecule* tab features an interactive chart panel that reveals the time evolution of biomolecule properties and provides region and position highlighting tools. To ensure image-derived measurements of biomolecule properties are evaluated together with primary observations, interactive image views linked to molecule records are supported through BDV integration (*Pietzsch et al., 2015*).

The Mars Rover was written to provide extensive possibilities for customization. All tabs and panels are defined in interfaces, independent of implementation details. This facilitates extension of the Mars Rover to support custom icons and display elements based on Molecule Archive type. This will enable further refinement and the development of workflow-specific displays by extending the core architecture in the future.

## Commands for image processing and biomolecule analysis

Mars comes with a collection of several dozen Fiji/ImageJ2 commands for common single-molecule image processing and analysis tasks (*Table 1*). This includes commands to find, fit, integrate, and track through time intensity peaks and objects in images. Commands to correct for non-uniform excitation beam profiles, and to transform region of interest peak collections with colocalization filtering possibilities. In addition to image processing, there are Molecule Archive commands for opening, merging, and transforming datasets as well as kinetic change point analysis. Finally, Mars has commands devoted to interoperability with other common formats. This includes importers for TrackMate (see the next section), single molecule dataset (SMD) (*Greenfeld et al., 2015*), and LUMICKS h5 files from optical tweezers experiments.

Commands appear in a submenu of the plugins menu of Fiji when Mars is installed using the update site. When a command is selected, users are presented with a dialog to choose the desired options. The active image in Fiji is used as input by image processing commands. Some provide preview possibilities prior to processing full image sequences. Tracking and fluorescence integration commands generate Molecule Archives as outputs which open in new windows when the command finishes. Biomolecules can then be explored using the Mars Rover user interface presented in all open Molecule Archive windows and all analysis work can be saved to disk as two files: one file with a

**Table 1.** Mars commands.

Description of Fiji/ImageJ2 commands supporting the analysis of image-derived biomolecule data in Mars. Detailed documentation can be found on the Mars documentation website (https://duderstadt-lab.github.io/mars-docs/).

| Command | Description |
| --- | --- |
| **Image** | |
| Peak Finder | Finds high-intensity pixel clusters (peaks) in an image. Additionally, the sub-pixel position can be determined utilizing a 2D Gaussian fit. |
| DNA Finder | Finds vertically aligned DNA molecules in an image. Additionally, the sub-pixel position of both ends of the molecule can be determined utilizing a 2D Gaussian fit. |
| Peak Tracker | Finds, fits, and tracks peaks in images. |
| Object Tracker | Identifies unspecified objects in images utilizing classification by segmentation and tracks their center of mass. |
| Molecule Integrator | Integrates the intensity of a peak over all frames. |
| Molecule Integrator (multiview) | Integrates the intensity of a peak over all frames in an image stack with multiview images. |
| Beam Profile Corrector | Corrects for the beam profile-generated image intensity deviations. |
| Gradient Calculator | Calculates the gradient of consecutive pixels from top to bottom or from left to right to identify long linear objects such as DNA molecules. |
| Overlay channels | Combines several individual videos into one creating a single video with the information stored along the 'Channel (C)' dimension. |
| **Molecule** | |
| Open Archive | Opens a Molecule Archive. |
| Open Virtual Store | Opens a virtual Molecule Archive. |
| Build Archive from Table | Converts an opened table with a 'molecule' index column into a Molecule Archive. |
| Build DNA Archive | Builds a DNA Molecule Archive from a single Molecule Archive and a list of DNA ROIs in the ROI Manager. It uses the location of the DNA molecules to search for molecules in the single Molecule Archive that overlap with (parts of) this location. |
| Merge Archives | Merges multiple Molecule Archives (placed in a single folder) into one. |
| Merge Virtual Stores | Merges multiple virtual Molecule Archives (placed in a single folder) into one. |
| Add Time | Adds a column to the molecule tables to convert time points (T) to real time values as specified in the metadata or by a user-defined time increment. |
| Drift Corrector | Calculates and corrects for the sample drift given a Molecule Archive and a tag corresponding to all immobile molecules in the dataset. Generates new columns for each molecule table. |
| Region Difference Calculator | Calculates the difference between the regions specified for all molecules in the Molecule Archive and adds the outcome as a molecule parameter. |
| Variance Calculator | Calculates the variance on a specified molecule table column and adds the outcome as a molecule parameter. |
| **Table** | |
| Open Table | Imports a comma or tab-delimited table to the MarsTable format. |
| Sort | Sorts a MarsTable based on values in a specified column. |
| Filter | Filters the rows of a MarsTable based on the specified criteria. |
| Import IJ1 Table | Imports any ImageJ1 table to the MarsTable format. |
| Import TableDisplay | Imports any SciJava table to the MarsTable format. |
| **KCP** | |
| Change Point Finder | Detects linear regions or steps in single-molecule traces. This command generates molecule segments tables listing endpoints and fits for linear regions. |

*Table 1 continued on next page*

*Table 1 continued*

| Command | Description |
| --- | --- |
| Single Change Point Finder | Detects a single change point in a single-molecule trace. The output is a segments table with the end points and fit or the position. |
| Sigma Calculator | Calculates the error value in a specific region of interest in all single-molecule traces that can be used as input for the change point calculation commands. |
| **ROI** | |
| Transform ROIs | Transforms peak ROIs from one region of a multiview image to another. |
| **Import** | |
| LUMICKS h5 | Opens optical tweezer data in HDF5 (h5 file extension) format collected using a LUMICKS instrument and converts the data to Molecule Archive format. |
| Single-molecule dataset (SMD) | Opens SMD files in plaintext json format and converts the data to Molecule Archive format. |

yama extension containing the primary Molecule Archive data and a second file with the yama.rover extension containing Mars Rover settings used to restore the window state and dashboard widgets. The typical Mars workflow starts with the creation of a Molecule Archive by processing an image sequence, followed by biomolecule feature analysis and tagging. This typically involves the creation of custom analysis scripts in Fiji that manipulate Molecule Archives. Final publication quality figures are often rendered by opening the yama file in a Jupyter notebook and using one of the Python charting libraries. Comprehensive reference materials and detailed step-by-step guides for common workflows are freely available on the Mars documentation website.

The UI-agnostic format of ImageJ2 commands ensures they can be used in many different contexts with or without a graphical user interface available. To facilitate these applications, methods have been added for all required settings and example scripts for commands. Commands can be combined into larger scripts that support multistep analysis workflows runnable on high-performance computing clusters. This facilitates a smooth transition from a dialog-based workflow development phase using the graphical user interface of Fiji to high-performance parallel processing of many experiments in environments lacking a graphical user interface.

## TrackMate interoperability

Fiji provides a comprehensive open-source platform for scientific image analysis containing well-established software for common imaging processing tasks. These technologies are integrated in a modular fashion as plugins that can be combined in limitless combinations. Mars commands and data structures are fully integrated into Fiji, simplifying interoperability with these technologies. The applications presented below provide examples of how Fiji plugins can be combined with Mars commands. To illustrate how this interoperability can be further extended, we developed an action to export TrackMate results to Molecule Archive format (Source code available at https://github.com/duderstadt-lab/mars-trackmate). TrackMate is a Fiji plugin for single-particle tracking that offers several tracking algorithms with a powerful user interface with many spot filtering and track editing tools (*Tinevez et al., 2017*). The action we developed adds an export option in the final TrackMate panel called 'go to Mars' that opens a Molecule Archive with the converted results. This feature is installed with Mars and requires no additional configuration. The Mars *Peak Tracker* is ideal for everyday single-molecule tracking problems with a few simple options in a single dialog but does not offer all the capabilities of TrackMate. This extension gives users more possibilities for complex problems such as tracking the shape and position of objects using machine learning algorithms (*Ershov et al., 2021*) and exporting the results from TrackMate to a Mars ObjectArchive.

## Applications

To demonstrate the range of analysis tasks that can be performed with Mars, we have developed three workflows based on real-world applications. In the first workflow, we determine the rate of transcription of single RNA polymerases by tracking their position as a function of time on long DNAs based on *Scherr et al., 2022*. In the second workflow, we demonstrate how to accurately analyze

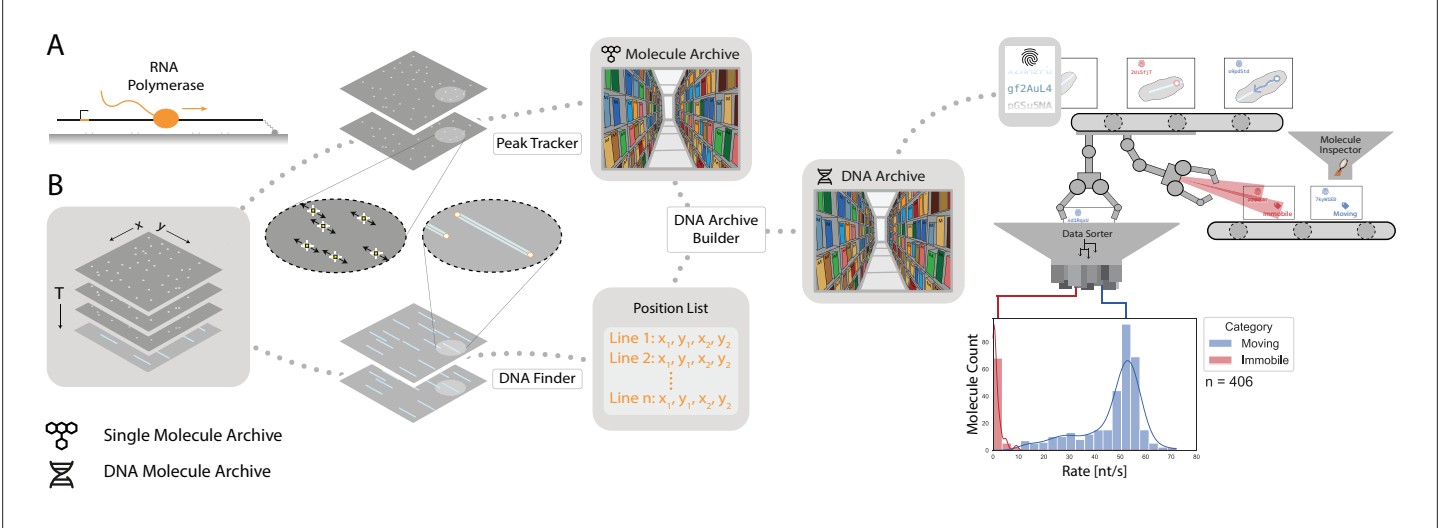

**Figure 3.** Workflow for tracking RNA polymerase position during transcription. (**A**) Schematic of the RNA polymerase assay. Promoter-containing surface-immobilized 21 kb DNA was incubated with fluorescently-labeled RNA polymerase after which transcription was tracked over time. (**B**) Representation of the analysis pathway showing the analysis steps starting from the raw image stack on the left to a final plot on the right. First, the *Peak Tracker* extracted position vs time information from each fluorescent RNA polymerase creating a single Molecule Archive. In parallel, the DNA finder located the long, line-shaped, DNA molecules and generates a list of start and end positions. The information yielded from both tools was merged into a final DNA Molecule Archive. A classification and sorting process was applied resulting in a final plot showing the abundance of tracked molecules at various transcription rates (nt/s). A Gaussian fit to the population with rates > 40 nt/s revealed a population average transcription rate of 53±3.6 nt/s. Here n is the number of molecules.

images containing observations of single-molecule FRET based on *Hellenkamp et al., 2018* and dynamic FRET arising from conformational switching of Holiday junctions (*Hyeon et al., 2012*). And finally, we highlight the virtual storage capability of Mars working with very large datasets from high-throughput imaging of DNA-tethered microspheres manipulated with forces and torques based on *Agarwal and Duderstadt, 2020*. Detailed step-by-step instructions for each workflow can be found on the Mars documentation website and the raw data used in each workflow are freely available on either GitHub or Zenodo. Jupyter notebooks, sample archives, and scripts used in the workflows are available in the mars-tutorials repository.

## Workflow 1–tracking RNA polymerase position during transcription

Single-molecule total internal reflection fluorescence (smTIRF) microscopy has become an indispensable tool to study biomacromolecular structure and functionality allowing the observation of, e.g., molecule position and dynamics. Examples of such studies include kinetic studies of DNA replication (*Dequeker et al., 2022*; *Ha et al., 2002*; *Lewis et al., 2020*), studies of the polymerization of structural elements like actin (*Amann and Pollard, 2001*), the direct observation of flagellar motor rotation (*Sowa et al., 2005*), as well as tracking of processes in vivo (*Vizcay-Barrena et al., 2011*). To illustrate the use of Mars for the analysis of such datasets, this example shows a typical Mars workflow for smTIRF studies of the kinetics of a fluorescently-labeled RNA polymerase transcribing on an immobilized, promoter-containing, 21 kb DNA molecule (*Scherr et al., 2022*; *Figure 3A*). In the presence of all four nucleotides, RNA polymerase could initiate transcription from the promoter and progress on the DNA which could be temporally and spatially visualized by measuring fluorescent emission upon excitation. After transcription was completed, DNA was poststained with SYTOX orange to reveal the position of the DNA molecules in the last frames of the video. By correlating the RNA polymerase movement with the position of the DNA molecule, information about the polymerase processivity and progression rates were obtained.

To analyze the data quantitatively, first, a beam profile correction was applied to remove the non-uniform laser excitation in the field of view due to the Gaussian beam profile of frequently employed light sources in TIRF microscopy. The *Peak Tracker* (*Figure 3B*) then determined the location of each fluorescent spot throughout the progressing frames and stored this information in a Single Molecule

Archive. The identified positions were subsequently corrected for sample drift, that occurred over the course of the measurement, with the *Drift Corrector*. In parallel, the last frames of the video, showing the DNA molecules, were fitted with the *DNA Finder* to generate a coordinate list with the positions of all DNA molecules. This information was correlated with the positional information of the polymerases and a DNA Molecule Archive was created. The obtained DNA Molecule Archive contains the positional information of all polymerase molecules found to be on the DNA and serves as the basis for further kinetic studies. Distinguishing between sub-populations is possible by sorting the molecule records either by means of parameter values and/or assigned molecule tags. Concluding the analysis, data exploration revealed a population-specific rate distribution (*Figure 3B*, right) showing an observed transcription rate of 53±3.6 nt/s which is well in line with previous studies reporting transcription rates between 40–80 nt/s (*Thomen et al., 2008*).

## Workflow 2–measuring intramolecular distances with smFRET

Single-molecule Förster Resonance Energy Transfer (smFRET) microscopy is used extensively to study protein dynamics (*Lerner et al., 2018*; *Mazal and Haran, 2019*; *Michalet et al., 2006*), RNA (*Seidel and Dekker, 2007*; *Shaw et al., 2014*; *Xiaowei, 2005*) and DNA (*Seidel and Dekker, 2007*) interactions, to elucidate enzyme mechanisms (*Smiley and Hammes, 2006*) as well as to study protein structure (*Dimura et al., 2016*; *Schuler and Eaton, 2008*) and molecular machines (*Hildebrandt et al., 2014*; *Stein et al., 2011*). Mars comes with multi-color fluorescence integration commands well-suited for the analysis of such datasets. Here we present a typical Mars workflow for dual-color smFRET data collected using alternating laser excitation with TIRF microscopy. This workflow starts with integration of fluorescence intensities and performs all stages of analysis to obtain corrected FRET efficiency (E) and stoichiometry (S) distributions that provide intramolecular distance information

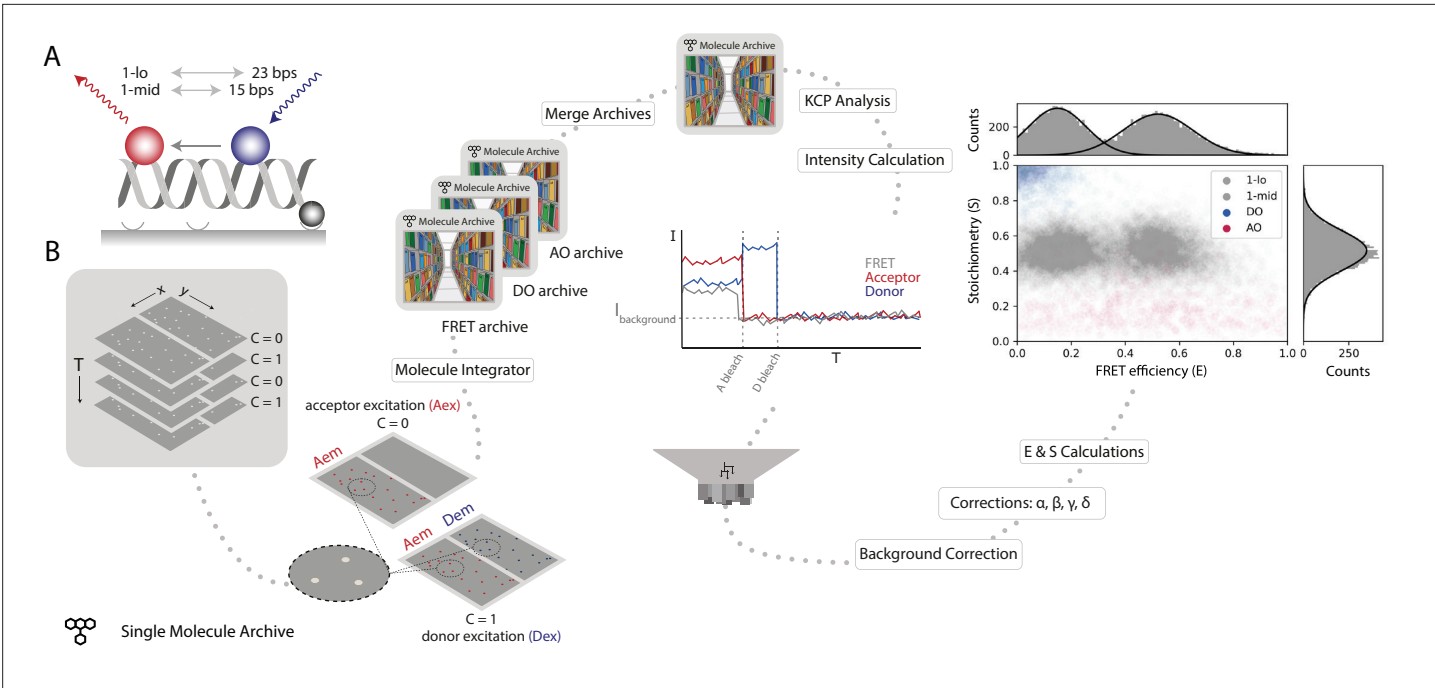

**Figure 4.** Workflow for a static smFRET experiment. (**A**) Schematic of the FRET assay. The FRET efficiency between two coupled dyes (donor, shown in blue, and acceptor, shown in red) on a short, immobilized, dsDNA oligo was measured. The inter-fluorophore distance was probed for two constructs: 23 bps (1-lo) or 15 bps (1-mid). (**B**) Representation of the analysis pathway starting from the raw image stack on the left to a stoichiometry vs FRET efficiency plot on the far right. First, the molecule integrator is used to integrate intensity vs time traces for each molecule (Aex: acceptor excitation, Aem: acceptor emission, Dem: donor emission, Dex: donor excitation) resulting in three Single Molecule Archives (FRET archive, donor only [DO] archive, and acceptor only [AO] archive). After merging, the data are corrected for background and other photo-physical effectsre classified according to the observed molecular features. Finally, the single-molecule data is displayed in a scatterplot with the stoichiometry (**S**) and FRET efficiency (**E**) information for both FRET samples (1-lo and 1-mid) as well as the AO and DO populations. The accompanying histograms plot the data from the 1-lo and 1-mid populations in gray bars and corresponding population-specific Gaussian fits as a solid black line. A detailed step-by-step guide to this workflow is available on the Mars documentation website.

and the kinetics of the molecule dynamics. The workflow is applied to both static and dynamic FRET datasets and was adapted to several collection strategies.

To illustrate how Mars is used to analyze static smFRET populations, we processed the publicly available dataset from the benchmark study of *Hellenkamp et al., 2018* to determine the FRET efficiency (E) and stoichiometry (S) of two short dsDNA-based samples labeled with donor (Atto550) and acceptor (Atto647N) fluorophores at two different inter-fluorophore distances (15 bps [1-lo] and 23 bps [1-mid], *Figure 4A*). Data were gathered by recording the emission from the surface-attached DNA molecules upon alternating donor and acceptor excitation (ALEX) to ensure accurate FRET measurements. Images were analyzed as follows, the *Peak Finder* was used to identify the locations of DNA molecules, followed by integration of the fluorescence intensity using the molecule integrator (multiview) to generate Molecule Archives with intensity vs time (I vs T) information (*Figure 4B*). To allow for the calculation of all correction factors, three Single Molecule Archives were generated: (i) a FRET archive including all DNA molecules that were found to fluoresce in both emission regions, (ii) an acceptor only (AO) archive containing all molecules that only have fluorescent acceptors excited with direct acceptor excitation, and (iii) a donor only (DO) archive containing all molecules that only exhibit donor emission. Separating these three species at the start of analysis facilitated easier downstream processing and data correction calculations. All three Molecule Archives were tagged and merged to a single master Molecule Archive containing the information of all three DNA molecule population types before further data corrections were applied.

First, a position-specific excitation correction was applied normalizing donor and acceptor intensities in relation to their specific position in the field of view. A subsequent kinetic change point analysis procedure (*Hill et al., 2018*) identified all intensity transitions associated with bleaching events. Next, molecules exhibiting all the expected smFRET features were selected. For example, molecules were not selected if they had more or less than exactly one donor and acceptor, displayed large intensity fluctuations not caused by the bleaching event, or low signal. Molecules were background corrected by subtracting the mean background intensity after bleaching from the measured intensity at each time point of the respective molecule. Furthermore, corrections accounting for leakage of donor emission into the acceptor region, quantum yield normalization, and direct excitation of the acceptor were applied to obtain accurate FRET parameters (see Methods for details). Finally, the corrected

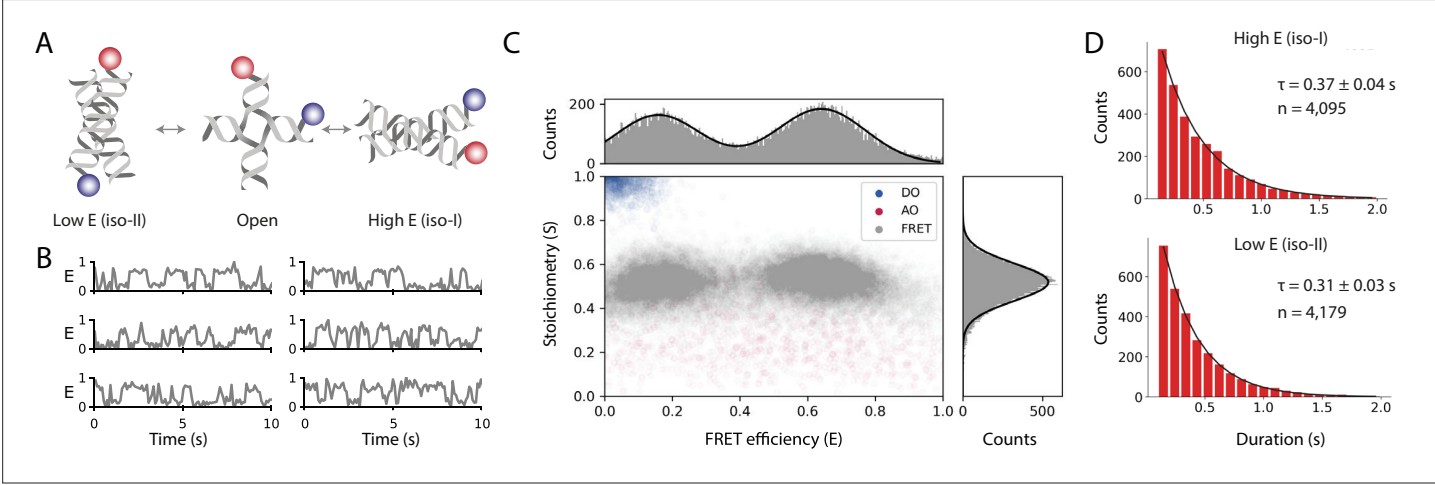

**Figure 5.** Workflow results for dynamic smFRET. (**A**) Schematic of the dynamic FRET substrate. The FRET efficiency between two dyes (donor, shown in blue, and acceptor, shown in red) attached to the arms of a Holliday junction exhibiting rapid interconversion between low and high FRET states was measured. Biotin attachment to the surface used during the experiment was omitted from the cartoon for clarity. (**B**) FRET efficiency as a function of time for representative molecules. (**C**) Scatterplot showing the stoichiometry vs FRET efficiency for all timepoints of accepted molecules for FRET, AO, and DO populations. One-dimensional histograms displayed along each axis are fitted with single or double Gaussian models for stoichiometry and FRET efficiency, respectively. (**D**) Dwell time distributions from a two-state model for the high and low FRET states. Time scales are from exponential fits with standard deviation and n is the number of dwells taken from 601 molecules. A detailed step-by-step guide to this workflow is available on the Mars documentation website.

The online version of this article includes the following figure supplement(s) for figure 5:

**Figure supplement 1.** Validation of dynamic smFRET.

traces allowed for the calculation of E and S values for each molecule. The average corrected E values obtained for 1-lo and 1-mid were $E_{1-lo}$ = 0.16 ± 0.11 and $E_{1-mid}$ = 0.53 ± 0.13 (mean ± SD) with 246 and 250 accepted molecules included in the final calculations, respectively. These values are in agreement with the results of the multilab study by Hellenkamp et al. ($E_{1-lo}$ = 0.15 ± 0.02 and $E_{1-mid}$ = 0.56 ± 0.03). The differences in precision are due to differences in error estimation. We report the standard deviation resulting from Gaussian fitting of each population. Whereas, Hellenkamp et al. have reported the standard deviation from the best estimates from many labs. When calculated with the standard error of the mean our estimates become $E_{1-lo}$ = 0.16 ± 0.01 and $E_{1-mid}$ = 0.53 ± 0.01.

Our analysis of the static FRET dataset illustrates how Mars can be used to determine the FRET efficiency (E) and stoichiometry (S) of fixed FRET populations and allowed for direct benchmarking against published values. However, the majority of smFRET studies typically contain transitions between FRET states resulting from distance changes during the observation time within each molecule. To illustrate how this workflow can be used to analyze samples that exhibit transitions between states, we imaged surface-immobilized Holliday junctions labeled with donor and acceptor fluorophores attached to different DNA arms with ALEX (*Figure 5A*). To enhance conformational switching of the arms, we introduced a buffer containing 50 mM magnesium as previously described (*Hyeon et al., 2012*). We applied the smFRET workflow described above for the static FRET case to obtain the corrected FRET efficiency and stoichiometry as a function of time. Frequent transitions between high and low FRET states are observed from Holliday junctions switching between iso-I and iso-II conformations (*Figure 5B*). A scatterplot of stoichiometry vs FRET efficiency for all accepted molecules has two well-defined populations (*Figure 5C*) which we fit with a double Gaussian distribution to obtain $E_{iso-I}$=0.16 ± 0.12 and $E_{iso-II}$=0.64 ± 0.13 (mean ± SD, for n=601 molecules). Finally, we determined the dwell time distributions using a simple threshold-based two state model from which we obtained the transition timescales $\tau_{iso-I}$ = 0.31 ± 0.03 seconds and $\tau_{iso-II}$ = 0.37 ± 0.04 seconds (mean ± standard error) (*Figure 5D*). We used this simple model based on our prior knowledge that the system exhibited two state behavior. Alternatively, when the number of states is not known and transitions occur on timescales longer than the imaging rate, the *Change Point Finder* can be used to identify transitions in an unbiased manner. Alternatively, Molecule Archives can be opened in Python and other kinetic analysis methods can be used, such as Hidden Markov Modeling.

The FRET datasets presented thus far demonstrate the advantages of the ALEX collection strategy. Alternating excitation offers acceptor emission information from direct excitation that is used to calculate the stoichiometry of the dyes in each molecule and exclude traces with fluctuations in the acceptor signal that are independent of FRET. This ensures a more robust determination of correction factors and greater reliability. However, it comes at a cost when imaging with smTIRF. The FRET sampling rate is lower and the observation time is reduced due to limited emission from the acceptor before photobleaching. As a consequence, there are good reasons to omit the acceptor excitation pulses when considering the FRET collection strategy. Therefore, we have developed a third smFRET workflow with the Holliday junction dataset but without using the direct acceptor excitation information to calculate the correction factors. A detailed step-by-step guide to this workflow is available on the Mars documentation website. In this case, the stoichiometry is not calculated due to the lack of direct acceptor information.

Molecule selection is one of the most important steps during smFRET analysis and also the one most susceptible to bias. Therefore, a consistent set of criteria must be adhered to throughout the selection process and the final set of accepted molecules should be evaluated against known smFRET features. These procedures ensure molecules with improper labeling, dye fluctuations independent of FRET, and other photophysical artifacts are removed from the analysis. Jupyter notebooks that generate an automated validation report for all smFRET examples are included in the mars-tutorials repository. In the report, the stability of the sum of fluorescence signals is evaluated using the coefficient of variation, the anti-correlation of donor and acceptor emission is quantified using the Pearson correlation coefficient, and the median values of stoichiometry and FRET efficiency are compared with expected values for different regions of FRET traces, among other validation tests. The results of these evaluations for the dynamic FRET dataset are displayed in *Figure 5—figure supplement 1* with suggested rejection thresholds. This provides an extra layer to improve the quality of final datasets and provides additional quantitative criteria to reduce bias.

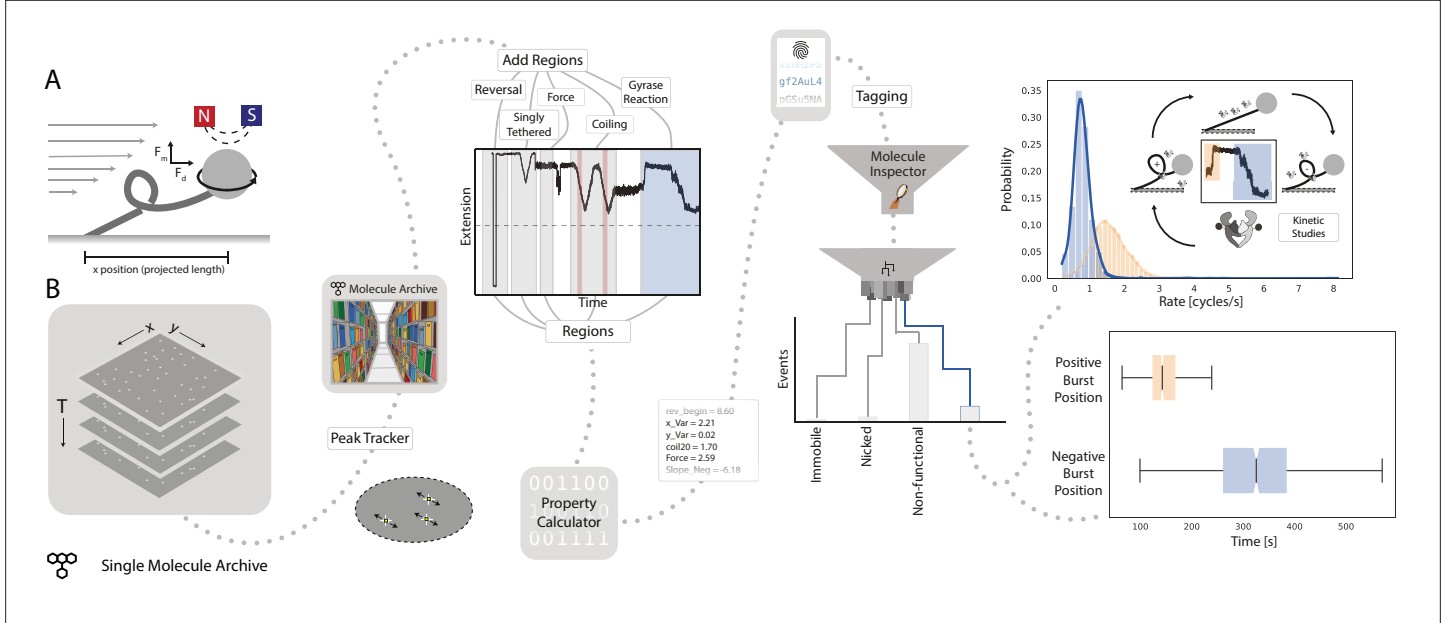

**Figure 6.** Workflow for gyrase characterization using flow magnetic tweezers. (**A**) Schematic of flow magnetic tweezers (FMT). The projected length of the surface-immobilized DNA molecule attached to a magnetic bead was measured under different flow as well as magnet height and rotation conditions to study changes in DNA topology. (**B**) Representation of the analysis workflow starting from the raw image stack on the left to the fully analyzed plots on the right. First, positional information is extracted by the *Peak Tracker* to yield a Single Molecule Archive. Regions assigned to specific parts of the experiment are highlighted in the example trace ('reversal', 'singly tethered', 'force', 'coiling', and 'gyrase reaction') and are used to calculate different DNA-related properties and parameters. Subsequent classification and tagging allows for easy exploration of subpopulations. The top graph shows the rate distribution (enzymatic cycles/s) found for gyrase activity resolving positive supercoils (orange) and introducing negative supercoils (blue), respectively. The lower graph shows a box plot of the delay between the introduction of the enzyme to the system (T=0) and the observed enzymatic activity. Plots were calculated from 2,406 individual molecules.

## Workflow 3–characterizing the kinetics of DNA topology transformations

Force spectroscopy methods, utilizing the high precision tracking of beads attached to biomolecules, have led to great insights into biological processes. Magnetic tweezers, in particular, are routinely used to study the physical behavior of DNA (*Nomidis et al., 2017*; *Strick et al., 1998*) as well as nucleic acid transformations by essential types of cellular machinery, such as structural maintenance of chromosomes complexes (*Eeftens et al., 2017*), the DNA replication machinery (*Burnham et al., 2019*; *Hodeib et al., 2016*; *Manosas et al., 2012*; *Seol et al., 2016*) and topoisomerases (*Charvin et al., 2005*; *Gore et al., 2006*; *Nöllmann et al., 2007*; *Strick et al., 2000*). The throughput of this approach has recently been improved by combining magnetic tweezers with DNA flow stretching to create an instrument called flow magnetic tweezers (FMT) (*Agarwal and Duderstadt, 2020*). The very large datasets generated by this instrument pose a challenge for analysis and initially led to the development of Mars. To illustrate the advantages offered by Mars in the analysis of these types of data, we present a workflow studying the behavior of DNA gyrase, a topoisomerase from *E. coli* (*Nöllmann et al., 2007*). This workflow illustrates the analysis steps required from raw tracking data to a well-structured single-molecule dataset with a traceable processing history utilizing the virtual storage infrastructure of Mars.

In the experimental setup of FMT, surface-immobilized DNA molecules are attached to paramagnetic polystyrene beads in a flow cell (*Figure 6A*). Block magnets placed above the flow cell create a magnetic field that orients the beads and applies a vertical force. A constant flow through the flow cell results in a drag force on the beads and the DNAs. Similar to a conventional magnetic tweezers setup, rotation of the magnets will change the topology of the DNA according to the direction of rotation. Furthermore, by inverting the flow direction, the DNA tether will flip and reveal the location of surface attachment. A low magnification telecentric lens makes it possible to image a massive field of view providing high throughput observations. At low applied forces, changes in DNA topology induced

through magnet rotation result in the formation of DNA supercoils and DNA compaction. This results in a decrease in the projected length of the DNA molecule observable as bead motion in the image. Despite the low magnification of the lens required for high throughput, subpixel fitting allows for high resolution tracking of DNA length over time. The tracking results are rich in information that allows for detailed molecule classification and quantification of enzyme kinetics.

In the experiment presented in *Figure 6B*, first, a series of predetermined flow and magnet transformations were executed to check tether quality. Then gyrase was introduced, resolving positive supercoils and subsequently introducing negative supercoils. Regions were assigned to the trace in which parameters were calculated and tags were allocated to the molecules based on these parameter values. In this particular example, four final categories discriminated molecules that were either retained for analysis, immobile, nicked, or rejected for various other reasons (e.g. attachment of multiple DNA molecules to one magnetic bead). The molecules retained for analysis were background corrected and the rate of positive relaxation and negative introduction was calculated and plotted (*Figure 6B*, right). A positive burst rate of 1.59 cycles/s, as well as a negative burst rate of 0.80 cycles/s, were found in this experiment. These values are in agreement with those reported previously (*Nöllmann et al., 2007*).

This workflow illustrates how Mars can be used to analyze large datasets virtually with on-the-fly data retrieval. Careful documentation is even more critical for large datasets where many stages of analysis must be entirely automated. The documentation framework provided in Mars ensures that the analysis is done reproducibly by entering each step in the log of the metadata record. This ensures a fully reversible workflow in combination with keeping all observations in the Molecule Archive, including those rejected from the final analysis. Mars can be of value for other major force spectroscopy methods, even though many do rely on z-directional movement instead of x-y tracking. Several options for external initial image processing are available from which results could be imported to Mars either in tabular form or using a scripting environment. Furthermore, this example proves that Mars is very flexible and can be used for any camera-based data where single-molecules are localized or tracked. Subsequently, Mars provides a platform with improved classification options, documentation, and data reusability for downstream analysis. The expandability of Mars will allow diversification of the workflow to integrate a broader range of force spectroscopy input data, for example, from optical and traditional magnetic tweezers.

## Discussion

Rapid improvements in bioimaging technologies have led to powerful new approaches to follow the time evolution of complex biological systems with unprecedented spatial and temporal resolution generating vast information-rich datasets. These observations must be efficiently and reproducibly analyzed, classified, and shared to realize the full potential of recent technological advances and ensure new biological phenomena are discovered and faithfully quantified. Single-molecule imaging approaches, in particular, have moved from obscurity in specialized physics laboratories to the forefront of molecular biology research. Nevertheless, surprisingly few reporting standards and common formats exist. While significant progress has been made in establishing common file formats for single-molecule FRET datasets that can store raw photon information from point detectors (*Greenfeld et al., 2015*) and time-binned trajectories from images (*Ingargiola et al., 2016*), they offer limited options for adaption to other experimental modalities. Mars provides a solution to bridge this gap in the form of a common set of commands for single-molecule image processing, a graphical user interface for molecule exploration, and a Molecule Archive file format for flexible storage and reuse of image-derived datasets adaptable to a broad range of experiment types. To ensure Mars is accessible to a large community, Mars is developed open source and freely available as a collection of SciJava commands distributed through a Fiji update site. The Mars project is a member of the Scientific Community Image Forum (*Rueden et al., 2019*) which provides a vibrant platform for new users to get help and for advanced users to find solutions to difficult image analysis problems.

The workflows described illustrate how the basic collection of modular commands and Molecule Archive transformations provided can be reused to analyze data from very different experimental configurations. Nevertheless, we recognize the commands are ultimately limited and will not address all problems easily. Therefore, we provide interoperability between Mars and other platforms available in Fiji. In particular, to provide access to a broader range of particle tracking options,

results can be exported from TrackMate to Molecule Archive format. We plan to expand interoperability to include other common formats generated from single-molecule imaging experiments as they emerge. Additionally, Mars uses the SCIFIO framework (*Hiner et al., 2016*) to convert different image formats into OME format and includes a specialized image reader for more comprehensive support of images recorded using Micromanager (*Edelstein et al., 2010*) frequently used for single-molecule imaging. This will allow Mars to process new image formats as they are developed and SCIFIO or Bio-Formats readers are written (*Linkert et al., 2010*). These are only a few examples of the many workflow options that integrate with core Fiji technologies. In case these integrations do not provide a solution, Mars has several built-in extension mechanisms. New Molecule Archive types and custom user interface elements can be added separately and discovered by Mars at runtime using the SciJava discovery mechanism. Mars was written with script and command development in mind to allow for the analysis of observations resulting from new approaches beyond single-molecule applications by extending the existing framework. Finally, Molecule Archives can easily be opened in Python environments by directly loading them as JSON files or using PyImageJ (*Curtis Rueden et al., 2021*), which provides access to many other data manipulation and visualization libraries.

Single-molecule imaging approaches have gained widespread usage and have become an indispensable tool for the discovery of new biological mechanisms. Unfortunately, data reporting standards have lagged far behind. Raw images together with a record of the processing history for reproduction are rarely provided. The development of new formats like the Molecule Archive format presented here will make it easier for researchers to faithfully report their results and ensure reproducibility. This will increase the level of confidence and quantitative accuracy of findings and allow for broader reuse of existing information-rich datasets. However, Mars only provides a framework to aid reproducibility and does not ensure it. Individual researchers are ultimately responsible for maintaining reporting standards sufficient for reproduction and following best practice recommendations for single molecule imaging experiments (*Lerner et al., 2021*). For example, scripts developed with Mars should be version controlled, made publicly accessible, and report all essential parameters to the processing log. Moreover, the version numbers and settings of all software integrated into workflows must be documented and reported to ensure reproducibility.

The discovery of new biological phenomena from single-molecule observations often depends on time-consuming manual classification of individual molecules and behaviors. Machine learning algorithms are now offering the possibility to automate these tasks (*Kapadia et al., 2021*; *Thomsen et al., 2020*), but their accuracy depends on robust training datasets. The powerful record tagging tools provided with Mars provide the ideal platform for the creation of large training datasets for machine learning based classifiers. Future work will focus on further development of interoperability of Mars with other platforms and machine learning workflows.

## Methods
### Mars installation

To get Mars, start by downloading a copy of Fiji and installing it. Open Fiji, go to the help menu, select update… and click the manage update sites button and activate the Mars update site by checking the box where you find Mars in the list of available update sites. Apply all required changes. This should install a large number of jar files that includes all core Mars software components and all dependencies required. To complete the installation process, quit and reopen Fiji. Now you are ready to start using Mars by running the commands in the Plugins menu in the Mars submenu. We suggest installing Mars into a new copy of Fiji to avoid incompatibility issues with older copies of Fiji. We will ensure Mars works with future copies of Fiji using the installation procedure outlined.

### Getting help with Mars

Mars is a community partner on the Scientific Community Image Forum (*Rueden et al., 2019*) where users can report their problems in posts with the mars tag to get feedback and troubleshooting support. Don't be a stranger! If you have a problem, no matter how small, we would love to hear about it and try to help. Solving your problem will likely help other users.

## Workflow 1–tracking RNA polymerase position during transcription

Specific details about protein purification and labeling, the microscope set-up, sample preparation, and the imaging procedure can be found in the publication by *Scherr et al., 2022*. The raw data accompanying this example workflow is freely available through the Mars tutorials GitHub repository .

Extensive background information about all described Mars commands, specific settings used, and screenshots of example Molecule Archives and algorithm outcomes can be found on the Mars documentation pages. Scripts and example Molecule Archives accompanying this analysis can be found in the Mars tutorials GitHub repository.

## Workflow 2–measuring intramolecular distances with smFRET

In this workflow, parameter nomenclature, as well as data correction and calculation procedures, were performed as described by *Hellenkamp et al., 2018* to facilitate comparisons and to illustrate the flexibility of Mars workflow creation. Specifics on sample design, preparation, and the data acquisition procedure for the static FRET dataset presented in *Figure 4* can be found in the original work (*Hellenkamp, 2018*). The raw image data is freely available on Zenodo. Dynamic FRET from conformational changes of a Holliday junction labeled with donor and acceptor dyes was collected for this study and deposited on Zenodo, where it is freely available for download. Full experimental details can be found below. Detailed step-by-step instructions for three smFRET workflows are available on the Mars documentation website for static FRET, dynamic FRET , and dynamic FRET without direct acceptor excitation . In addition to the detailed step-by-step guides on the workflow documentation pages linked above, we have also created a YouTube channel with many tutorials and included a video showing how to perform the dynamic FRET workflow.

The most significant difference between the three examples is the method used to calculate gamma ($\gamma$), the normalization of effective fluorescence quantum yields, and detection efficiencies of the acceptor and donor. For datasets collected using ALEX, the $\gamma$ correction factor is calculated using a linear fit of one over the stoichiometry vs FRET efficiency as described previously (*Lee et al., 2005*). In the case of static FRET, this is conducted with the combined 1-lo and 1-mid datasets. For dynamic FRET, the two populations within each molecule are separated and fit. In the absence of direct acceptor excitation, $\gamma$ is calculated using the ratio of the changes in intensities before and after acceptor photobleaching as described previously (*McCann et al., 2010*).

The analysis starts with the identification of the fluorescent DNA molecule positions and subsequent intensity vs time trace extractions. The Mars image processing commands used in the smFRET workflow were written for videos where different excitation wavelengths are stored in different channels which is standard practice throughout the imaging community. However, the static FRET sample data (https://doi.org/10.5281/zenodo.1249497) was saved as a single image sequence without channel information despite alternating acceptor and donor excitation for each frame. Therefore, we wrote a script for Fiji to convert to the multichannel format expected by Mars that de-interleaves the frames into two channels: acceptor excitation (C=0) and donor excitation (C=1). The intensity peaks were identified using the *Peak Finder* and their locations are exported to the ImageJ ROI Manager. To account for the dual-view of the camera which separates the acceptor emission and donor emission wavelengths to different regions, the coordinates of the peaks were transformed to the other half of the dual-view using an Affine2D matrix. This matrix was calculated with the bead data provided on Zenodo using the 'descriptor-based registration (2d/3d) (*Preibisch et al., 2010*) plugin in Fiji as described in the tutorial on the Mars documentation site . Next, the molecule integrator *(multiview)* was used to extract the intensity vs time traces of all molecules at all specified emission and excitation colors. This generated a single Molecule Archive containing molecule records with intensity traces for each identified molecule. For easier downstream analysis, the described procedure was repeated, thereby generating three different Molecule Archives from each video: (i) FRET archive–a Molecule Archive containing molecules that have both donor and acceptor emission; (ii) AO archive–a Molecule Archive containing molecules with acceptor emission after acceptor excitation only; and (iii) DO archive–a Molecule Archive containing molecules with only donor emission after donor excitation. The metadata records of these Molecule Archives were tagged accordingly before merging them into a master Molecule Archive using the *Merge Archives* command.

In the master Molecule Archive, after the metadata tags (FRET, AO, and DO) were added to all molecule records , a position-specific excitation correction was applied that normalizes the donor and

acceptor intensities dependent on their location in the field of view . Next, for each intensity trace, the *Single Change Point Finder* was used to automatically detect large intensity shifts indicating donor or acceptor bleaching events . Subsequent manual tagging of molecule traces with the 'accepted' tag allowed for the exclusion of certain molecules in further analysis: (i) molecules with either more than or fewer than exactly one donor and acceptor fluorophore per molecule, respectively, (ii) molecules showing large intensity changes other than the bleaching event, and (iii) molecules with a too low signal to noise ratio, for example. This yielded a Molecule Archive with tagged molecules to be included in the forthcoming calculations and data correction steps.

Next, the fluorescence emission from the donor and acceptor were corrected and the FRET efficiency and stoichiometry values were calculated. All steps were combined in one script . The corrections and calculations performed by this script are described in the following steps, subscripts, superscripts, and variables are defined as follows:

- i-iii: (i) the uncorrected intensity; (ii) intensity after background correction; (iii) intensity after background, $\alpha$ and $\delta$ corrections.
- D or A: donor or acceptor.
- Aem|Dex: intensity in the acceptor channel upon donor excitation.
- Dem|Dex: intensity in the donor channel upon donor excitation.
- Aem|Aex: intensity in the acceptor channel upon acceptor excitation.
- app: apparent values that include systematic experimental offsets.
- DO/AO: donor-only/acceptor-only species.
- S: stoichiometry, approximately the ratio of the donor dyes to all dyes.
- E: FRET efficiency.
- $I_{emission|excitation}$: intensity for the specified excitation and emission.
- $F_A|D$: acceptor emission upon donor excitation fully corrected for background, leakage, and direct excitation.
- $F_D|D$: donor emission upon donor excitation fully corrected for background, and differences in fluorescence quantum yields and detection efficiencies of the acceptor and donor.
- $F_A|A$: acceptor emission upon acceptor excitation fully corrected for background, and differences in excitation intensities and cross-sections of the acceptor and donor.
- $\alpha$: leakage of donor fluorescence into the acceptor region.
- $\delta$: direct acceptor excitation by the donor excitation laser.
- $\beta$: normalization of excitation intensities and cross-sections of the acceptor and donor.
- $\gamma$: normalization of effective fluorescence quantum yields and detection efficiencies of the acceptor and donor.

Step 1: The background correction step subtracts the mean background intensity after bleaching as measured from the traces in the respective FRET Molecule Archive from all prebleaching intensities in a trace-wise fashion. This yielded the background-corrected intensity values ($^{ii}I_{emission|excitation}$). These corrected intensity values were used to calculate $^{ii}E_{app}$ and $^{ii}S_{app}$ (**Equation 1**).

$$^{ii}S_{app} = \frac{^{ii}I_{Aem|Dex} + {}^{ii}I_{Dem|Dex}}{^{ii}I_{Aem|Dex} + {}^{ii}I_{Dem|Dex} + {}^{ii}I_{Aem|Aex}} \; and \; {}^{ii}E_{app} = \frac{^{ii}I_{Aem|Dex}}{^{ii}I_{Aem|Dex} + {}^{ii}I_{Dem|Dex}} \tag{1}$$

Step 2–3: Next, the leakage of donor fluorescence into the acceptor channel ($\alpha$) and direct acceptor excitation by the donor excitation laser ($\delta$) were calculated from the AO and DO molecules respectively according to (**Equation 2**). $F_{A|D}$ stores the fully corrected intensity from acceptor emission upon donor excitation for each molecule at each time point before the first photobleaching event (**Equation 3**).

$$\alpha = \frac{\left\langle {}^{ii}E_{app}^{(DO)} \right\rangle}{1 - \left\langle {}^{ii}E_{app}^{(DO)} \right\rangle} \; and \; \delta = \frac{\left\langle {}^{ii}S_{app}^{(AO)} \right\rangle}{1 - \left\langle {}^{ii}S_{app}^{(AO)} \right\rangle} \tag{2}$$

$$F_{A|D} = {}^{ii}I_{Aem|Dex} - \alpha {}^{ii}I_{Dem|Dex} - \delta {}^{ii}I_{Aem|Aex} \tag{3}$$

Step 4–5: To then account for the normalization of excitation intensities and cross-sections of the acceptor and donor ($\beta$) and the normalization of effective fluorescence quantum yields and detection efficiencies of the acceptor and donor ($\gamma$) the respective correction factors were determined based on the relationship between the 1-lo and 1-mid population. To do so, $^{ii}E_{app}$ and $^{ii}S_{app}$ values were averaged in a molecule-wise fashion and linear regression against the values of the entire population yielded correction factors $\beta$ and $\gamma$ (**Equation 4**) that were applied to calculate $F_{A|A}$ and $F_{D|D}$ (**Equation 5**).

Where $F_{A|A}$ is the corrected acceptor emission upon acceptor excitation and $F_{D|D}$ is the corrected donor emission on donor excitation. The gamma correction factor was calculated based on the assumption that placing the same dyes at different bases along the DNA molecule does not influence the ratio of the acceptor and donor fluorescence quantum yields. These subsequently yielded the fully corrected E and S parameters for each molecule (*Equation 6*). A population-specific molecule average revealed the respective population average values.

$$\frac{1}{\left\langle {}^{iii}S_{app}^{(FRET)}\right\rangle} = b * \left\langle {}^{iii}E_{app}^{(FRET)}\right\rangle + a \tag{4}$$

where $\qquad \beta = a + b - 1 \qquad$ and $\qquad \gamma = \frac{a-1}{a+b-1}$

$$F_{D|D} = \gamma *{}^{ii}I_{Dem|Dex} \text{ and } F_{A|A} = \frac{1}{\beta}*{}^{ii}I_{Aem|Aex} \tag{5}$$

$$E = \frac{F_{A|D}}{F_{D|D}+F_{A|D}} \text{ and } S = \frac{F_{A|D}+F_{D|D}}{F_{D|D}+F_{A|D}+F_{A|A}} \tag{6}$$

More information regarding the derivation of the discussed formula as well as information about the applied corrections can be found in the publication by *Lee et al., 2005*. We developed a second workflow using the Holliday junction dataset but without using the direct acceptor excitation information for corrections and calculations . In the absence of direct acceptor excitation, γ is calculated using the ratio of the changes in intensities before and after acceptor photobleaching as described previously (*McCann et al., 2010*).

$$\gamma = \frac{{}^{A}I_{Pre} - {}^{A}I_{Post}}{{}^{D}I_{Post} - {}^{D}I_{Pre}} \tag{7}$$

where ${}^{A}I_{Pre}$ is the mean acceptor intensity before photobleaching, ${}^{A}I_{Post}$ is the mean background of the acceptor spot after photobleaching, ${}^{D}I_{Pre}$ is the mean intensity of the donor before acceptor photobleaching, and ${}^{D}I_{Post}$ is the mean intensity of the donor after acceptor photobleaching before donor photobleaching (the donor recovery period).

The dwell time distributions for the dynamic FRET workflow examples using the Holliday junction were determined using a simple two-state model run on the final archive that adds segments tables to all molecule records . The segment tables with the state fits from the script are used in the Jupyter notebook to calculate the timescales of conformational changes displayed in *Figure 5*.

To assess the quality of the selected molecules in the final Molecule Archives, we developed several validation criteria and included validation reports for all three smFRET examples that can be found in the FRET section of mars-tutorials repository within the subfolder for each example workflow. In the report, the stability of the sum of fluorescence signals is evaluated using the coefficient of variation, anti-correlation of donor and acceptor emission is quantified using the Pearson correlation coefficient, and the median values of stoichiometry and FRET efficiency are compared with expected values for different regions of FRET traces, among other validation tests. The results of these evaluations for the dynamic FRET dataset are displayed in *Figure 5—figure supplement 1* with suggested rejection thresholds. The Jupyter notebooks provided only report on these validation parameters. We also provide a script to filter the selected molecules based on thresholds that can be used to automate part of the selection process . We offer this as a starting point, but care must be taken when performing the selection process and we expect additional criteria may need to be added to ensure robust automated filtering.

Furthermore, information about all described Mars commands, specific settings used in the built-in tools, and screenshots of expected outcomes can be found on the Mars documentation pages. Scripts, Jupyter notebooks, and Molecule Archives accompanying this analysis can be found in the Mars tutorials GitHub repository.

## Dynamic smFRET data collection

Dynamic single-molecule FRET datasets were obtained using a four-stranded holliday junction assembled as previously described (*Hyeon et al., 2012*). Briefly, HPLC purified oligos R_branch_bio (5′-biotin-TTTTTTTTCCCACCGCTCG<u>GCTCAACTGGG</u>-3′), H_branch_Cy3 (5′-Cy3-<u>CCGTAGCAGCG</u> CGAGCGGTGGG-3′), X_branch (5′-GGGCGGCGACCT CCCAGTTGAGCGCTTGCTAGGG-3′), and

B_branch_Alexa647 (5'-Alexa647-CCCTAGCAAGCCGCTGCTACGG-3') obtained from Eurofins Genomics GmbH were mixed to a final concentration of 10 µM each in annealing buffer (30 mM Hepes pH 7.5, 100 mM potassium acetate), heated to 90°C, and annealed by slow cooling to 4°C over 90 min. Imaging was performed using an RM21 micromirror TIRF microscope from Mad City Labs (MCL, Madison Wisconsin, USA) with custom modifications as previously described (*Larson et al., 2014*) equipped with an Apo N TIRF 60 × oil-immersion objective (NA 1.49, Olympus). Dyes were excited with OBIS 532 nm LS 120 mW and 637 nm LX 100 mW lasers from Coherent at full power, expanded to fill the field of view. Scattered light from excitation was removed and signals were separated with emission filter sets (ET520/40 m and ZET532/640 m, Chroma). Emission light was split at 635 nm (T635lpxr, Chroma) with an OptoSplit II dualview (Cairn Research, UK) and collected on an iXon Ultra 888 EMCCD camera (Andor).

Imaging surfaces were prepared as follows. Glass coverslips (22×22 mm, Marienfeld) were cleaned with a Zepto plasma cleaner (Diener Electronic) and incubated in acetone containing 2% (v/v) 3-aminopropyltriethoxysilane for 5 min. Silanized coverslips were rinsed with ddH$_2$O, dried, and baked at 110°C for 30 min. Coverslips were then covered with a fresh solution of 0.4% (w/v) Biotin-PEG-Succinimidyl Carbonate (MW 5,000) and 15% (w/v) mPEG-Succinimidyl Carbonate (MW 5,000) in fresh 0.1 M NaHCO$_3$ and incubated overnight at room temperature. Coverslips were rinsed with ddH$_2$O, dried, and incubated again with a fresh Biotin-PEG/mPEG solution as described above. Functionalized PEG-Biotin microscope slides were again washed and dried and finally stored under vacuum.

A functionalized PEG-Biotin microscope slide was covered with 0.2 mg/ml streptavidin in blocking buffer (20 mM Tris-HCl, pH 7.5, 50 mM NaCl, 2 mM EDTA, 0.2 mg/ml BSA, 0.005% (v/v) Tween20) for 15 min. Flow cells were assembled using ~0.1 mm thick double-sided tape containing an ~0.5 mm wide flow lane. The double-sided tape was sandwiched between the cover glass on one side and a 1 mm thick piece of glass on the other containing entry and exit holds. Polyethylene tubes (inner diameter 0.58 mm) were inserted into the holes and the entire assembly was sealed with epoxy.

Flow cells were flushed with blocking buffer and left for 30 min. Holliday junctions were incubated in the flow chamber at a concentration of 1–5 pM for 2 min in TN Buffer (10 mM Tris pH 8.0, 50 mM NaCl). Excess holliday junctions were removed by washing with TN buffer. Finally, imaging was performed in RXN Buffer (10 mM Tris pH 8.0, 50 mM MgCl$_2$, 1 mM Trolox (aged for 5–8 min under UV), 2.5 mM PCA and 0.21 U/ml PCD). Tubes were sealed with clips several minutes prior to imaging to allow time for oxygen removal. Before exciting the dyes in each field of view, an 808 nm laser was used to obtain focus. Collection was performed using an ALEX approach in which the 532 and 637 lasers were alternated and 40 ms exposures were collected in burst acquisition mode using a custom Bean-Shell collection script with micromanager 2.0. Ten to twenty fields of view were collected sequentially for 3 min each using an automated microDrive stage from MCL with autofocus steps preceding each collection. Image sequences are available on Zenodo (https://doi.org/10.5281/zenodo.6659531).

## Workflow 3–characterizing the kinetics of DNA topology transformations

The raw video is available through Zenodo (https://zenodo.org/record/3786442#.YTns2C2B1R0). This video is a reduced dataset from one of the FMT experiments investigating the topological changes gyrase performs on the DNA. The groovy scripts for analyzing the dataset can be found on GitHub . The first three scripts have been used to create a CSV file and the data was then plotted with a Python script. The data analysis procedure using Mars has been described in *Agarwal and Duderstadt, 2020*.

After tracking the molecules in the dataset and the generation of the single Molecule Archive, the traces were sorted and classified. The experiment was designed in such a way that steps in the process of the assay could be used as indicators. Based on these indicators, a discrimination was made between the molecules classifying them either as immobile (not mobile, not coilable), nicked (mobile, not coilable) or fit for analysis (mobile, coilable). A fourth category contains every rejected molecule due to various reasons like multiple DNA attachments on a single bead or getting stuck during the experiment before the enzymatic activity was detected. The tethers can appear to be mobile and coilable but certain thresholds are not passed like being coilable at high force. Specifically, the main indicators for this dataset are derived from: (i) the observation of a positional change of the DNA-bead after flow direction reversal (stuck or not stuck), (ii) a test for DNA molecule coilability by rotating the magnets at high force (single or multiple DNA molecules attached to a single bead), (iii) a molecular

force measurement investigating the force on the DNA, and (iv) a second rotating step at lower force to introduce positive twist which is resolved by gyrase. The positive and negative introduction of supercoiling is used to calibrate the extension change of each DNA molecule to the change of twist during the analysis. The response of the molecule to these indicator tests is determined by parameter calculation scripts that only consider a certain assigned region of interest in the trace. When a certain threshold for a parameter is met, a tag is added to the molecule record. The final set of tags present in the molecule record determines in which group the molecule is classified. A sliding window approach is applied to all relevant molecules revealing kinetic information such as coiling and compaction rates as well as influences on these parameters after the introduction of gyrase. The size of the window depends on whether gyrase is relaxing positive twist or introducing negative twist. For the positive relaxation a window of 12.5 s and for the negative introduction a window of 25 s is used. In this case, one cycle means the linking number of the DNA is changed by –2. Since the DNA has to be relaxed to get negatively supercoiled, the burst position for positive relaxation comes earlier than the negative introduction. The time points have been corrected such that gyrase introduction coincides with T=0. The calculated information was either exported to CSV format or was directly interpreted with Python. This flexibility enables data visualization on various platforms.

The entire analysis, including more background information, can be found on the website. Furthermore, the Mars commands are explained in great detail on the documentation site (https://duderstadt-lab.github.io/mars-docs/docs/).

## Data availability

The analysis software described is publicly available in several repositories on GitHub at https://github.com/duderstadt-lab. The core library used for the analysis and storage of data is contained in the mars-core repository. The graphical user interface is contained in the Mars-fx repository. The videos used in all workflows have been made available in public databases. Links can be found in the methods sections for each workflow. Extensive documentation and links to many additional resources and scripts used in all workflows can be found on the Mars documentation website.

## Acknowledgements

We are grateful to Giovanni Cardone (MPIB imaging facility) and Evelyn Plötz for their helpful feedback on the project. We would like to thank Prof. Thorsten Hugel for providing the Igor source code used for the analysis of smFRET data (*Hellenkamp et al., 2018*). We would like to thank Andreas Walbrun and Prof. Matthias Rief for providing the sample data used to develop the Mars LUMICKS h5 importer. We are grateful for the recent development efforts in the ImageJ2 and Fiji communities that have greatly enhanced the applications across platforms. We would like to thank Curtis Rueden, Jan Eglinger, Tobias Pietzsch, and Jean-Yves Tinevez for taking the time to answer the many questions that have come up during development and considering our pull requests. We thank Curtis Rueden for helping to configure our repositories for deployment to SciJava maven. This work was funded by the Deutsche Forschungsgemeinschaft (DFG, German Research Foundation)–SFB863–11166240, a starting grant from the European Research Council (Project Number: 804098, REPLISOMEBYPASS), and the Max Planck Society.

## Additional information

### Funding

| Funder | Grant reference number | Author |
| --- | --- | --- |
| European Research Council | ERC-StG-804098 | Nadia M Huisjes<br>Karl E Duderstadt<br>Matthias J Scherr<br>Barbara Safaric |
| Deutsche Forschungsgemeinschaft | SFB863-11166240 | Thomas M Retzer<br>Rohit Agarwal |

| Funder | Grant reference number | Author |
|---|---|---|
| Max Planck Society | | Lional Rajappa<br>Anita Minnen<br>Karl E Duderstadt |

The funders had no role in study design, data collection and interpretation, or the decision to submit the work for publication.

## Author contributions

Nadia M Huisjes, Conceptualization, Software, Formal analysis, Investigation, Visualization, Writing - original draft, Writing – review and editing; Thomas M Retzer, Conceptualization, Software, Formal analysis, Investigation, Visualization, Methodology, Writing - original draft, Writing – review and editing; Matthias J Scherr, Conceptualization, Data curation, Software, Investigation, Visualization, Writing – review and editing; Rohit Agarwal, Data curation, Software, Formal analysis, Methodology, Writing – review and editing; Lional Rajappa, Resources, Investigation; Barbara Safaric, Validation, Visualization, Writing – review and editing; Anita Minnen, Conceptualization, Methodology, Writing – review and editing; Karl E Duderstadt, Conceptualization, Resources, Data curation, Software, Formal analysis, Supervision, Funding acquisition, Validation, Investigation, Visualization, Methodology, Writing - original draft, Project administration, Writing – review and editing

## Author ORCIDs

Nadia M Huisjes http://orcid.org/0000-0002-4182-7229
Thomas M Retzer http://orcid.org/0000-0002-7353-2708
Matthias J Scherr http://orcid.org/0000-0003-0110-3709
Karl E Duderstadt http://orcid.org/0000-0002-1279-7841

## Decision letter and Author response

Decision letter https://doi.org/10.7554/eLife.75899.sa1
Author response https://doi.org/10.7554/eLife.75899.sa2

# Additional files

## Supplementary files

• Transparent reporting form

## Data availability

The analysis software described is publicly available in several repositories on GitHub at https://github.com/duderstadt-lab. The core library used for the analysis and storage of data is contained in the mars-core repository. The graphical user interface is contained in the mars-fx repository. The videos used in all workflows have been made available in public databases. Links to datasets can be found in the Methods section for each workflow. Extensive documentation and links to many additional resources and scripts used in all workflows can be found at https://duderstadt-lab.github.io/mars-docs/.

The following previously published datasets were used:

| Author(s) | Year | Dataset title | Dataset URL | Database and Identifier |
|---|---|---|---|---|
| Hugel T | 2018 | Datasets for "Precision and accuracy of single-molecule FRET measurements - a multi-laboratory benchmark study" | https://doi.org/10.5281/zenodo.1249497 | Zenodo, 10.5281/zenodo.1249497 |
| Agarwal R, Duderstadt KE | 2020 | Flow Magnetic Tweezers example video | https://zenodo.org/record/3786442#.YwbNPHMLJc | Zenodo, 10.5281/zenodo.3786442 |

*Continued on next page*

*Continued*

| Author(s) | Year | Dataset title | Dataset URL | Database and Identifier |
|---|---|---|---|---|
| Duderstadt KE | 2022 | Dynamic FRET example videos related to "Mars, a molecule archive suite for reproducible analysis and reporting of single-molecule properties from bioimages" | https://doi.org/10.5281/zenodo.6659531 | Zenodo, 10.5281/zenodo.6659531 |

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
