## [Editor Report]

This is a valuable paper that reports an open-source platform for the storage and processing of single-molecule, camera-based, imaging data. The development and testing of the platform are very compelling and the platform will facilitate data sharing and reproducibility and will be of great interest to practitioners of single-molecule imaging experiments, both experienced and new to the field. The work represents significant and important steps towards unifying and standardizing how the field stores and processes data and expanding the base of researchers who can easily employ single-molecule imaging methods.

---

## [Decision Letter]

**Decision letter after peer review:**

Thank you for submitting your article "Mars, a molecule archive suite for reproducible analysis and reporting of single molecule properties from bioimages" for consideration by *eLife*. Your article has been reviewed by 2 peer reviewers, and the evaluation has been overseen by Ruben Gonzalez as the Reviewing Editor and Volker Dötsch as the Senior Editor. The following individuals involved in review of your submission have agreed to reveal their identity: Eitan Lerner (Reviewer #1); Jingyi Fei (Reviewer #2).

Essential revisions:

1) The authors should further develop the Mars platform to: (i) expand the flexibility of the smFRET input data files in the manner described by Reviewer 2 and, if necessary, to do the same for the other various types of data files (e.g., single-molecule force-extension data from laser tweezer data, particle tracking data from DNA curtain experiments, etc.) and (ii) include validation information/reports of the kind described by Reviewer 1.

2) The authors should comprehensively test and troubleshoot all of the functions of the platform using different computers and operating systems to ensure the robust performance of the platform. Reviewer 2 and her student have kindly provided a detailed, but not exhaustive, list of technical problems that they encountered and that should be resolved.

3) The sample dataset of smFRET trajectories from Hellenkamp et al., (2017) that is analyzed for the work reported in this manuscript is limited in the sense that the trajectories do not exhibit transitions between different FRET efficiency states as a function of time. Given that the majority of smFRET studies typically contain trajectories that do exhibit such transitions, it is important that the authors include a second sample dataset of smFRET trajectories that do exhibit such transitions as a function of time. The authors are encouraged to analyze an already existing dataset from their own laboratory or a collaborator's laboratory or, if that is somehow not possible, to record such a dataset de novo. When adding these new analyses and revising the smFRET section of the manuscript, the authors should take care to also address Reviewer 1's comments regarding the smFRET example(s).

4) The authors should revise the manuscript to include: (i) a beginners-level, practical discussion of why a platform such as Mars is needed to increase the reproducibility of data storage and processing practices (including examples of real-world problems that Mars would address, minimize, or eliminate) and (ii) an easy-to-follow, brief guide explaining the use of Mars within the manuscript aimed at readers who may not necessarily be ready to use the platform or the associated Jupyter notebooks and/or interactive features of the website.

5) The step-by-step instructions for using Mars that are included on the website should be expanded to include further details, particularly in terms of the output files, as well as to include a general troubleshooting guide for users.

*Reviewer #1 (Recommendations for the authors):*

This report constitutes the review of the manuscript titled "Mars, a molecule archive suite for reproducible analysis and reporting of single molecule properties from bioimages", by Huisjes, Retzer et al. With the ever increasing interest in these techniques, accompanied by ever increasing experimental schemes and analytical frameworks it becomes difficult to report such experimental results in a manner that is universal, one which can adapt to any type of experiment and analysis, one which assists in reproducibility, and one which will keep the data as compact as possible, yet its usage as efficient as possible. This work introduces a novel platform for the reproducible archiving of camera-based single-molecule imaging experiments. This work will appeal to practitioners of single-molecule imaging experiments, both experienced and newbies. The readers of this work would benefit from understanding how to employ a rational data archiving process using Mars, following three examples the authors provide, which exhibit the generality of the platform and its ease of use. Readers who might want to employ Mars for their own single-molecule imaging measurements can also experience an intuitive guide on Mars GitHub or in Jupyter Notebooks the authors provide. However, I believe that it would even be better if the authors could provide a guidance chapter in the manuscript itself, in addition and before sending the readers to the guide online – this way, readers would already have an idea of what to expect when they try constructing their own Mars data archiving.

I judge that to the most part, the manuscript shows concisely and clearly how to use Mars for proper archiving of single-molecule imaging data, in a rational manner, that can be easily read, well understood, without loss of information and with the ability to easily track back and perform changes. The manuscript is very well polished, the figures provide a self-explanatory graphical guide – overall I enjoyed reading it and see how much this tool can advance the field. The importance of this work is very clear to experienced practitioners, who already know what it takes to provide extensive reports on their results, after being analyzed, corrected, filtered and analyzed in one combination of ways out of many others. Although I am very positive about this manuscript, I have a few suggestions for the authors:

1. Readers of the manuscript who are not yet experienced with single-molecule imaging could benefit from a pragmatical discussion that will explain the need for Mars for reproducibility and/or readability of the data: the different types of filtrations a practitioner may perform, all are context dependent, and what might happen if those are not properly documented and annotated. I suggest providing user examples of what might go wrong, and how Mars minimizes such cases.

2. The authors provide a very nice interactive as well as notebook guidance for readers of the manuscript who might want to get acquainted with Mars. However, I think readers of the manuscript would benefit from such a guide already in the manuscript, which will assist them in understanding what to expect when they move to try using Mars. This way, readers who are not yet ready to test Mars, would already be able to follow the guide in text, and then decide if they move to test Mars in action for their datasets.

3. The analysis pipeline provides a nice abstraction all the way from the raw data to the filtered information. However, another layer of validation should be added afterwards. For example, procedures that test all the filtered smFRET data for features that should exist in the data: (1) sum of all donor-excited signals should be constant after corrections were applied; (2) sum of all intensities should be constant, after corrections were applied, (3) transitions between states (FRET dynamics, or photobleaching) should exhibit donor and acceptor anti-correlation; (4) transitions between Stoichiometry states should exhibit donor and acceptor anti-correlation. Adding a filtered data validation procedure will enhance the procedure even better, and provide another layer of credibility, perhaps even in an automated report as an outcome of the validation step applied on all molecules.

4. Comments on the smFRET example:

a. The authors chose to show the TIRF camera-based smFRET data from Hellenkamp et al., (2017), of a static mixture of two DNA FRET constructs with two different mean FRET efficiencies. The procedure they describe was designed to fit the treatment of the static mixture of two types of immobilized molecules. Therefore, the analysis pipeline employed the change point finder only for the identification of photo-bleaching steps, and not for identifying dynamic FRET transitions. While this might be fine for the static heterogeneity of this sample, for the analysis framework to be as generic as possible for analyzing smFRET measurements, it is important to include a step of identification of state dwells within each single molecule FRET trajectory. Importantly, the FRET calculations should be performed on dwells that have been proven to belong to a single state.

b. The analysis pipeline was designed to filter out traces with large intensity changes that are not due to photo-bleaching. Large intensity changes might be interesting if one of the FRET dyes experiences quantum-yield changes that are unrelated to FRET (e.g., smFRET involving Cy3 and Cy5). It could be interesting in the proper context, and hence should allow the user to change the data rejection criteria, in a context-dependent manner.

c. Correction factors 1: the text relies on the correction factors presented by the multi-lab work of Hellenkamp et al., (2017). That approach uses the Lee et al., (2005) approach for calculating the γ correction factor, which relies on the linear dependence of the inverse of the mean stoichiometry on the mean proximity ratio, of multiple single-population smFRET-ALEX measurements. While this correction factor procedure is used by many and is very common, it does suffer from assuming that both donor and acceptor fluorescence quantum yields change in the same manner when positioning the same donor and acceptor dyes at different positions along the same molecule (a dsDNA in this case). This might not be true, as different abelling positions introduce different dye microenvironments, which may introduce different quenching rates that are unrelated to FRET. In other words, it is possible (and not uncommon at all) that different molecules with different FRET values will also be associated with different γ factors. There are alternatives that can assist in calculating per-population γ factors, which require knowledge of the donor and acceptor fluorescence lifetimes per population. However, this is not yet possible in immobilized TIRF camera-based smFRET. Therefore, in page 10 of the bioRxiv preprint, when the authors present the γ correction factor, I suggest they add: “… , under the assumption that placing the same dyes at different bases along the DNA molecule does not influence the ratio of the acceptor and donor fluorescence quantum yields”.

d. Correction factors 2: the text relies on the correction factors presented by the multi-lab work of Hellenkamp et al., (2017). That approach uses the Lee et al., (2005) approach for calculating the γ correction factor. However, Hellenkamp et al., (2017) and not Lee et al., (2005) is cited. I ask the authors to cite Lee et al., (2005), when providing the explanation on the β- and γ-factor calculations.

e. When explaining the FRET-related calculations, sometimes the variables are not explained: (i) in steps 4 and 5, explicitly explain the readers what are FA|A and FD|D; (ii) "FA|D stores the fully corrected acceptor fluorescence intensity value"; (iii) explain for FRET novices what do the variables in equations 1 and 2 mean.

f. At the bottom of page 6 of the bioRxiv preprint, the authors provide their estimations of the FRET efficiency values, and it seems that the accuracy is high, if comparing the values to the mean values reported by Hellenkamp et al., (2017). However, the precision of the reported values (E1-lo = 0.14 {plus minus} 0.13 and E1-mid = 0.51 {plus minus} 0.09) are low compared to the precisions in Hellenlamp et al., (E1-lo = 0.15 {plus minus} 0.02 and E1-mid = 0.56 {plus minus} 0.03). Why such imprecisions? Please explain the readers in the text.

*Reviewer #2 (Recommendations for the authors):*

Technical issues encountered:

1. Overall the step-wise tutorial on the website is well organized, but can be further improved. Particularly, the expected results/output files/windows should also be included, similar as the parameter setting for the input part. Sometimes it’s unclear what the correct output format should be from each operation.

2. We noticed Mac and Windows systems could encounter different issues. Taking the FRET analysis as an example, we had to switch between the Windows and Mac systems to get it through completely, because certain steps worked in one but not the other. For example:

(1) When we tried to convert the example FRET video to contain channel information using script 1, the Windows system generated a blank image. However, it ran correctly on Mac.

(2) There was some problem with transforming the ROIs to the right part of the split view on a Mac system. We used the same parameters as the instruction, but all the coordinates disappeared after doing Transform ROIs. But it worked well on a windows system.

(3) On a Mac system, after running peak tracer or molecular integrator, we were not able to generate a GUI interface containing the actual output, but a console saying the analysis is successful.

3. It’s very confusing that the output file of the converted FRET data is a composite image of two channels. It is not explained what each of the channels is until explaining the output file of the molecular integrator step. Echoing my point 1, it is necessary to explain to the users the f’rmat of the output files. The composite image makes it appear that two channels are the s’me without any explanation, and each channel is the sum of the red and green signal.

---

## [Author Response]

Essential revisions:1) The authors should further develop the Mars platform to: (i) expand the flexibility of the smFRET input data files in the manner described by Reviewer 2 and, if necessary, to do the same for the other various types of data files (e.g., single-molecule force-extension data from laser tweezer data, particle tracking data from DNA curtain experiments, etc.) and (ii) include validation information/reports of the kind described by Reviewer 1.

(i) We have revised Mars to expand the flexibility of the smFRET input types as requested by Reviewer #2:

1. We added support for multiview setups having more than a dualview. The Transform ROIs and Molecule Integrator (multiview) commands now support tripleview, quadview, and more.

2. We added two new example smFRET workflows using a new dataset we collected that has dynamic FRET from a Holliday junction substrate. One example demonstrates a complete ALEX analysis workflow of dynamic smFRET data with alternating red and green pulses. (https://duderstadt-lab.github.io/mars-docs/examples/Dynamic_FRET/)

3. As requested by the reviewer, the second new example demonstrates smFRET analysis when there is no direct acceptor excitation. In this case, we provide new scripts that perform molecule-by-molecule γ calculation and added several other modifications for this type of input data. (https://duderstadt-lab.github.io/mars-docs/examples/No_aex_FRET/)

4. We have added instructions for processing smFRET data collection using setups with two different cameras.

Expand the flexibility of Mars input types to include single-molecule force-extension data from laser tweezer data, particle tracking from DNA curtain experiments:

1. We created a new repository called mars-lumicks (https://github.com/duderstadt-lab/mars-lumicks) with an import command for lumicks h5 optical tweezer datasets. This import command reads the h5 file created by the widely used lumicks optical tweezer setups and converts it into a Molecule Archive. The Molecule Archive with the converted data then opens directly in Fiji. This new import command is now included in the Mars update site that is part of normal Mars installations. Details instructions on use can be found in the repository readme.

2. Many performance enhancements were required in mars-fx (the Mars GUI we call the Mars Rover) to nicely handle the converted lumicks datasets, which had substantially higher time resolution than typical mars datasets. For example, one dataset has 10s of millions of points, so a new down sampler was added to allow seamless zooming in and out. Memory handling was also improved to better support these datasets.

3. We created a new repository called mars-smd (https://github.com/duderstadt-lab/mars-smd) with an import command for the Single-molecule Dataset (SMD) format. This format was proposed for single-molecule FRET datasets. The importer was tested on the example data provided in the original data format specification. We failed to locate newer datasets with this format for further testing. We hope users will get in touch if they have this data type and would like to use Mars.

4. The Mars Peak Tracker supports particle tracking from DNA curtain experiments. The tracking example provided on the mars-docs website can be used for these datasets, the only difference compared to the sample data provided is the more compact organization of DNA molecules in the DNA curtains.

Reviewer #2 also asked how Mars could handle raw data in different formats. For example, with red and green channels as separate videos. As a start, we improved the script we wrote for conversion of the Hellenkamp *et al.* Nature Methods (2018) dataset that adds the channel information Mars expects to the raw video (https://github.com/duderstadt-lab/mars-tutorials/blob/master/Example_workflows/FRET/scripts/static_FRET_reformat_video.groovy). In addition, we tested Mars with more video formats. Mars can support all formats that can be opened in Fiji. This includes the hundreds of formats supported by Bio-Formats. We also previously wrote a micro-manager 1.4 and 2.0 reader that is included with Mars and nicely extracts all metadata from these kinds of files (https://github.com/duderstadt-lab/mars-scifio). This includes the new Holliday junction sample data we collected with micro-manager 2.0.

When writing Mars, we made a huge effort to fully comply with and support the OME format. We strongly feel that members of the single-molecule imaging community should make a greater effort to collect data with this common and universally understood format. However, we understand this is unlikely to happen anytime soon.

Nevertheless, we are confident that videos collected with any strategy can be converted to a format Mars understands due to the robustness of the OME format. For the specific case of separate red and green videos mentioned by Reviewer#2, both videos could be opened in Fiji and combined as different channels using the Merge Channels ImageJ command (Image>Color>Merge Channels). The scripts and instructions included with Mars can help in conversion of other video types. However, without specific sample data to use for testing, we can only guess about the many different methods people could use for collecting their data when they decide to ignore common image format standards generally accepted in the microscopy community. We are very happy to provide further conversion scripts that accept other collection types we have not yet seen. Mars is a community partner on the Scientific Community Image Forum. This is the perfect place for users to post about different video input types and we can help provide conversion scripts.

We have also added a general discussion of supported image formats on the website in the documentation section under ‘Supported Image Formats’ to further address this comment (https://duderstadt-lab.github.io/mars-docs/docs/). This includes all types supported by Fiji, but some workflows depend on using multiple channels that separate excitation/filter differences. We provide instructions for converting where required.

(ii) Include validation information/reports of the kind described by Reviewer #1:

We have added validation information and a jupyter notebook validation report based on the suggestions from Reviewer #1 to the smFRET example workflows and scripts. In particular, we now calculate the coefficient of variation of the sum of the donor and acceptor signals and sum of all signals to check for the expected constant fluorescence. We calculate the Pearsons correlation coefficient in the FRET region to check for anti-correlation of the donor and acceptor fluorescence. We check that the E and S values match expected theoretical values for distinct regions of the FRET trace. These metrics have been added by the smFRET scripts as molecule parameters and are then used in the jupyter notebook validation report. The report notebook can be used with any dataset processed with the smFRET scripts we provide. We have generated sample records for all three smFRET workflows based on the sample Molecule Archives we provide in the mars-tutorials repository:

dynamic FRET example:

https://github.com/duderstadt-lab/mars-tutorials/blob/master/Example_workflows/FRET/dynamic/dynamic_FRET_validation_report.ipynb

dynamic FRET example no acceptor excitation:

https://github.com/duderstadt-lab/mars-tutorials/blob/master/Example_workflows/FRET/no_acceptor_excitation/no_acceptor_excitation_FRET_validation_report.ipynb

static FRET example:

https://github.com/duderstadt-lab/mars-tutorials/blob/master/Example_workflows/FRET/static/static_FRET_validation_report.ipynb

All these notebooks render nicely when viewed in Firefox on the github website without the need to install and run them locally (we noticed safari doesn’t render them as nicely due to issues with the nbviewer developed on github). They can be run with other datasets just by changing the files paths at the top.

We have included a new Figure 5 —figure supplement 1 with the results of validation for the new Holliday junction dataset.

These many new features and performance enhancements have greatly improved Mars. We were happy the editors and reviewers asked for these changes.

2) The authors should comprehensively test and troubleshoot all of the functions of the platform using different computers and operating systems to ensure the robust performance of the platform. Reviewer 2 and her student have kindly provided a detailed, but not exhaustive, list of technical problems that they encountered and that should be resolved.

We are very grateful to Reviewer #2 and her student for reporting the issues they encountered. We were very sad to hear about these problems. We have resolved the technical problems they encountered. In the process, we have also completely remade the smFRET workflows to try and avoid further issues. This involved streamlining all the steps and creating a uniform set of scripts for all workflows. We also rewrote the static FRET conversion script that gave problems (https://github.com/duderstadt-lab/mars-tutorials/blob/master/Example_workflows/FRET/scripts/static_FRET_reformat_video.groovy). We have tested the workflows on mac and windows. A new member of our lab has also been testing mars on windows more extensively for several months.

To help further with these issues and make it clear what is expected for different commands, we have created a YouTube channel (https://www.youtube.com/channel/UCkkYodMAeotj0aYxjw87pBQ) that has video tutorials for working with Mars. Many are short tutorials, but we have also added an in-depth video going through the dynamic FRET example step-by-step (https://www.youtube.com/watch?v=JsyznI8APlQ). This addresses a comment by reviewer #1 asking for further information for users that are not yet ready to try Mars. In that case, they can now watch the videos to see exactly how it works in use. We hope these videos will also help with issues because steps can now be precisely followed.

Despite all our efforts we imagine there could be computer configurations we have not yet encountered and issues we haven’t encountered yet could occur. As mentioned above, Mars is a community partner on the Scientific Community Image Forum. This is the perfect place for users to post about problems they encounter or errors they see.

3) The sample dataset of smFRET trajectories from Hellenkamp et al., (2017) that is analyzed for the work reported in this manuscript is limited in the sense that the trajectories do not exhibit transitions between different FRET efficiency states as a function of time. Given that the majority of smFRET studies typically contain trajectories that do exhibit such transitions, it is important that the authors include a second sample dataset of smFRET trajectories that do exhibit such transitions as a function of time. The authors are encouraged to analyze an already existing dataset from their own laboratory or a collaborator's laboratory or, if that is somehow not possible, to record such a dataset de novo. When adding these new analyses and revising the smFRET section of the manuscript, the authors should take care to also address Reviewer 1's comments regarding the smFRET example(s).

This is an excellent suggestion. We recorded a new dataset de novo in our laboratory to address this point. We chose a Holliday junction substrate very similar to the one reported by Hyeon *et al.*, 2012, Nat. Chem. (https://www.nature.com/articles/nchem.1463) that exhibits dynamic changes between low FRET (iso-II) and high FRET (iso-I) conformations on the 100s of milliseconds timescale. We collected on many fields of view in ALEX format with our dualview microscope. We have deposited the new dataset on zenodo (https://doi.org/10.5281/zenodo.6659531) to make it publicly available. We developed two new smFRET example workflows using the new sample data:

(https://duderstadt-lab.github.io/mars-docs/examples/Dynamic_FRET/) and

(https://duderstadt-lab.github.io/mars-docs/examples/No_aex_FRET/). In the second one we ignore the direct acceptor excitation information to provide scripts needed to process datasets where that hasn’t been collected. We also provided a script for kinetic analysis of dwell time distributions and revised the smFRET workflow text of the manuscript accordingly to include a description of the newly collected dataset. We have included a new figure 5 showing the results of analysis of the new datasets and included the experimental details in the methods scripts of the revised manuscript.

4) The authors should revise the manuscript to include: (i) a beginners-level, practical discussion of why a platform such as Mars is needed to increase the reproducibility of data storage and processing practices (including examples of real-world problems that Mars would address, minimize, or eliminate) and

This is a great suggestion. In this revised manuscript, we have included a new section right after the Introduction called “Common pitfalls in single-molecule image-processing workflows” that provides a beginners-level, practical discussion of why Mars is needed and how it addresses real world reproducibility and data storage issues specific to single-molecule datasets.

(ii) an easy-to-follow, brief guide explaining the use of Mars within the manuscript aimed at readers who may not necessarily be ready to use the platform or the associated Jupyter notebooks and/or interactive features of the website.

We understand that many readers of our manuscript might not be ready to try mars and read through the other documentation and example materials we have created. For these readers, we see that a brief guide explaining the use of Mars would be helpful. We have expanded the ‘Commands for image processing and biomolecule analysis’ to include such a guide. There are limitless ways to combine the commands. We have also created a YouTube channel with tutorials and example workflow videos (https://www.youtube.com/channel/UCkkYodMAeotj0aYxjw87pBQ). We think this might be the very best way for people to quickly get a feeling for the mechanics of Mars.

5) The step-by-step instructions for using Mars that are included on the website should be expanded to include further details, particularly in terms of the output files, as well as to include a general troubleshooting guide for users.

Based on the feedback from the reviewers, we have extensively revised and updated the Mars documentation website. We have included a new FAQ page to help with troubleshooting and more extensive troubleshooting sections in the smFRET workflow examples. We also created a YouTube channel that has video tutorials and workflow examples to help clarify the expected inputs and outputs.

As mentioned above, it is very hard for us to anticipate all problems that could arise depending on the dataset used and computer configuration. The Scientific Community Image Forum is the very best place for many of these discussions. Forum posts are public and provide very helpful troubleshooting information for all users. Over time and more posts are made a much broader range of questions are addressed. We can only hope users will reach out for help there.

Reviewer #1 (Recommendations for the authors):This report constitutes the review of the manuscript titled "Mars, a molecule archive suite for reproducible analysis and reporting of single molecule properties from bioimages", by Huisjes, Retzer et al. With the ever increasing interest in these techniques, accompanied by ever increasing experimental schemes and analytical frameworks it becomes difficult to report such experimental results in a manner that is universal, one which can adapt to any type of experiment and analysis, one which assists in reproducibility, and one which will keep the data as compact as possible, yet its usage as efficient as possible. This work introduces a novel platform for the reproducible archiving of camera-based single-molecule imaging experiments. This work will appeal to practitioners of single-molecule imaging experiments, both experienced and newbies. The readers of this work would benefit from understanding how to employ a rational data archiving process using Mars, following three examples the authors provide, which exhibit the generality of the platform and its ease of use. Readers who might want to employ Mars for their own single-molecule imaging measurements can also experience an intuitive guide on Mars GitHub or in Jupyter Notebooks the authors provide. However, I believe that it would even be better if the authors could provide a guidance chapter in the manuscript itself, in addition and before sending the readers to the guide online – this way, readers would already have an idea of what to expect when they try constructing their own Mars data archiving.I judge that to the most part, the manuscript shows concisely and clearly how to use Mars for proper archiving of single-molecule imaging data, in a rational manner, that can be easily read, well understood, without loss of information and with the ability to easily track back and perform changes. The manuscript is very well polished, the figures provide a self-explanatory graphical guide – overall I enjoyed reading it and see how much this tool can advance the field. The importance of this work is very clear to experienced practitioners, who already know what it takes to provide extensive reports on their results, after being analyzed, corrected, filtered and analyzed in one combination of ways out of many others. Although I am very positive about this manuscript, I have a few suggestions for the authors:1. Readers of the manuscript who are not yet experienced with single-molecule imaging could benefit from a pragmatical discussion that will explain the need for Mars for reproducibility and/or readability of the data: the different types of filtrations a practitioner may perform, all are context dependent, and what might happen if those are not properly documented and annotated. I suggest providing user examples of what might go wrong, and how Mars minimizes such cases.

We are very happy the reviewer appreciates the central goal of Mars to create a common format that helps to enforce reproducible analysis workflows that can easily be shared with others. We very much appreciate the positive feedback on the work that has been a long-term dream of our lab.

In this revised manuscript, we have included a new section right after the Introduction called “Common pitfalls in single-molecule image-processing workflows” that provides a beginners-level, practical discussion of why Mars is needed and how it addresses real world reproducibility and data storage issues specific to single-molecule datasets. We kept the explanation as brief as possible while still addressing the comment. Several pages could be devoted to this topic, but we would surely lose the reader in the process.

We thank the reviewer for this suggestion. This improves the manuscript for the broader audience beyond single-molecule experts.

2. The authors provide a very nice interactive as well as notebook guidance for readers of the manuscript who might want to get acquainted with Mars. However, I think readers of the manuscript would benefit from such a guide already in the manuscript, which will assist them in understanding what to expect when they move to try using Mars. This way, readers who are not yet ready to test Mars, would already be able to follow the guide in text, and then decide if they move to test Mars in action for their datasets.

We understand that many readers of our manuscript might not be ready to try mars and read through the other documentation and example materials we have created. For these readers, we see that a brief guide explaining the use of Mars would be helpful. We have revised the “Commands for image processing and biomolecule analysis” section to include a brief guide that describes the use of Mars starting from using menus and dialogs. We have also created a YouTube channel with tutorials and example workflow videos (https://www.youtube.com/channel/UCkkYodMAeotj0aYxjw87pBQ). We think this might be the very best way for people to quickly get a feeling for the mechanics of Mars without using it themselves or looking at the example pages.

As noted by the reviewer, we have put tremendous effort into the Mars documentation website and materials included in the mars-tutorials repository. These serve as living documentation for Mars that goes far beyond the scope of what can be covered in the manuscript, which is focused on introducing the conceptual framework for Mars. Therefore, we have included this new section in the methods of the revised manuscript. We feel this is the most appropriate place for these details if they are to be included in the manuscript.

3. The analysis pipeline provides a nice abstraction all the way from the raw data to the filtered information. However, another layer of validation should be added afterwards. For example, procedures that test all the filtered smFRET data for features that should exist in the data:

This is a fantastic suggestion that we very much appreciate. We have revised the smFRET workflow scripts to include additional table columns and molecule parameters that provide validation information. We have also written a validation jupyter notebook that assess all the key criteria described below. We have included a new Figure 5 —figure supplement 1 with the validation results from the new dynamic smFRET dataset. We explain our validation calculations point-by-point below.

1) sum of all donor-excited signals should be constant after corrections were applied;

In the smFRET scripts used for calculating all the correction factors we have added a section that introduces a new table column called SUM_Dex that is the sum of the donor and acceptor signals. To assess whether it is constant, we tested many different metrics and finally setting on the coefficient of variation. We calculate the coefficient of variation for the FRET region and add this as a new molecule parameter called SUM_Dex_FRET_Coefficient_of_Variation. Valid traces seem to have values in the 0 to 0.4 region.

**Author response image 1. sa2fig1:** Histogram comparing the Coefficient of Variation of SUM_Dex for Accepted FRET molecules compared to all FRET molecules for the dynamic FRET example. Lower values reflect valid constant signal. The SUM_Dex can also be plotted and directly inspected using the Mars Rover.

2) sum of all intensities should be constant, after corrections were applied,

In the smFRET scripts used for calculating all the correction factors we have added a section that introduces a new table column called SUM_signal that is the sum of all signals (donor and acceptor emissions on donor excitation as well as acceptor emission on acceptor excitation). We calculate the coefficient of variation for the FRET region and add this as a new molecule parameter called SUM_signal_FRET_Coefficient_of_Variation. Valid traces seem to have values in the 0 to 0.3 region.

**Author response image 2. sa2fig2:** Histogram comparing the Coefficient of Variation of SUM_signal for Accepted FRET molecules compared to all FRET molecules for the dynamic FRET example. Lower values reflect valid constant signal. The SUM_signal can also be plotted and directly inspected using the Mars Rover.

3) transitions between states (FRET dynamics, or photobleaching) should exhibit donor and acceptor anti-correlation;

In the smFRET scripts used for calculating all the correction factors we have added a calculation of the Pearsons correlation coefficient (comparing donor and acceptor emission due to FRET with donor excitation). We calculate the Pearsons correlation coefficient for the FRET region and add this as a new molecule parameter called FRET_Pearsons_Correlation. We see strong anti-correlation for the dynamic FRET example in the range of -1 to -0.25.

**Author response image 3. sa2fig3:** Histogram comparing the Pearsons correlation coefficient for donor and acceptor emission during FRET for Accepted FRET molecules compared to all FRET molecules for the dynamic FRET example. Valid molecules should have strong anti-correlation as seen for the accepted molecules.

As expected, we don’t see anti-correlation for the static FRET example because there are no dynamic changes, so this metric is not useful in that case. We made a note of this in the corresponding validation notebook.

4) transitions between Stoichiometry states should exhibit donor and acceptor anti-correlation).

We have added a section in the validation notebooks that assess the expected values of S depending on the region of the FRET trace. Here we note that this is somewhat different for TIRF smFRET data as compared to smFRET data collected using in-solution FRET from burst see in a confocal microscope. We include the region between dye bleach events for this analysis.

**Author response image 4. sa2fig4:** Histograms comparing stoichiometry values for FRET, donor bleaches first, or acceptor bleaches first for Accepted FRET molecules compared to all FRET molecules for the dynamic FRET example. Regions in between dye bleach events are often very short leading to broader distributions.

Adding a filtered data validation procedure will enhance the procedure even better, and provide another layer of credibility, perhaps even in an automated report as an outcome of the validation step applied on all molecules.

The new table columns and molecule parameters are calculated and introduce by two scripts depending on the workflow used:

https://github.com/duderstadt-lab/mars-tutorials/blob/master/Example_workflows/FRET/scripts/FRET_workflow_4_alex_corrections.groovy

or

https://github.com/duderstadt-lab/mars-tutorials/blob/master/Example_workflows/FRET/scripts/FRET_workflow_6_corrections_without_aex.groovy

The molecule parameters are then used in the jupyter notebook validation reports. The report notebook can be used with any dataset processed with the smFRET scripts we provide. We have generated sample notebooks for all three smFRET workflows based on the sample Molecule Archives we provide in the mars-tutorials repository:

dynamic FRET example:

https://github.com/duderstadt-lab/mars-tutorials/blob/master/Example_workflows/FRET/dynamic/dynamic_FRET_validation_report.ipynb

dynamic FRET example no acceptor excitation:

https://github.com/duderstadt-lab/mars-tutorials/blob/master/Example_workflows/FRET/no_acceptor_excitation/no_acceptor_excitation_FRET_validation_report.ipynb

static FRET example:

https://github.com/duderstadt-lab/mars-tutorials/blob/master/Example_workflows/FRET/static/static_FRET_validation_report.ipynb

All these notebooks render nicely when viewed in Firefox on the github website without the need to install and run them locally (we noticed safari doesn’t render them as nicely use to issues with the nbviewer developed on github). They can be run with other datasets just by changing the files paths at the top.

Finally, these provide rough guesses about thresholds for filtering. However, we did not perform any filtering in our analysis workflow, but only plotted the results from our manual assessment of the traces. We include a final script that could be used to filter the data based on several of the criteria we assess in the validation steps. We provide this as a starting point for users that would like to pursue this strategy, but we feel extra care must be taken. We have made a note of this.

Validation filtering script:

https://github.com/duderstadt-lab/mars-tutorials/blob/master/Example_workflows/FRET/scripts/FRET_workflow_7_validation_filter.groovy

Since the initial submission, we have worked very hard to integrate Conda Python 3 support directly into Mars in Fiji without using a Jupyter notebook or switching platforms (https://forum.image.sc/t/fiji-conda/59618). A β version of this is now working. However, it requires launching Fiji from Python following these instructions (https://github.com/duderstadt-lab/marspylib). This will allow for validation reports to be generated directly in Molecule Archives open in Fiji using Python charting libraries. As part of this work the comments tab now supports multiple documents and rendering of python fenced code blocks. Also, python dashboard widgets are supported, so that Python charts be used directly in the Mars Rover. This is an ongoing collaboration with the PyImageJ/Fiji developers that is not get finished and has required the creation of several new repositories: (https://github.com/scijava/scripting-python) and (https://github.com/duderstadt-lab/marspylib). Sadly, this is beyond the scope of the current work, but will soon provide a much more powerful reporting toolkit.

We would like to thank the reviewer again for this excellent suggestion. We feel this greatly improved Mars and we learned a lot about smFRET analysis in the process.

4. Comments on the smFRET example:a. The authors chose to show the TIRF camera-based smFRET data from Hellenkamp et al., (2017), of a static mixture of two DNA FRET constructs with two different mean FRET efficiencies. The procedure they describe was designed to fit the treatment of the static mixture of two types of immobilized molecules. Therefore, the analysis pipeline employed the change point finder only for the identification of photo-bleaching steps, and not for identifying dynamic FRET transitions. While this might be fine for the static heterogeneity of this sample, for the analysis framework to be as generic as possible for analyzing smFRET measurements, it is important to include a step of identification of state dwells within each single molecule FRET trajectory. Importantly, the FRET calculations should be performed on dwells that have been proven to belong to a single state.

We agree! We recorded a new dataset de novo in our laboratory to address this point. We chose a Holliday junction substrate very similar to the one reported by Hyeon *et al.*, 2012, Nat. Chem. (https://www.nature.com/articles/nchem.1463) that exhibits dynamic changes between low FRET (iso-II) and high FRET (iso-I) conformations on the 100s of milliseconds timescale. We collected on many fields of view in ALEX format with our dualview microscope. We have deposited the new dataset on zenodo (https://doi.org/10.5281/zenodo.6659531) to make it publicly available. We developed two new smFRET example workflows using the new sample data:

(https://duderstadt-lab.github.io/mars-docs/examples/Dynamic_FRET/) and

(https://duderstadt-lab.github.io/mars-docs/examples/No_aex_FRET/). In the second one we ignore the direct acceptor excitation information to provide scripts needed to process datasets where that hasn’t been collected. We also provided a script for kinetic analysis of dwell time distributions (https://github.com/duderstadt-lab/mars-tutorials/blob/master/Example_workflows/FRET/scripts/FRET_workflow_5_two_state_dwell_times.groovy) and some suggestions for use of change point and other types of kinetic analysis.

In this case, we used a very simple two-state model based on a threshold to generate the dwell time distributions shown in panel D of the Figure 5. For multi-state FRET cases where the number and E values of the states are not known, and their lifetimes are longer than the sampling rate, the change point command included in mars is ideally suited. The notebooks provide an example of how to calculate dwell time distributions from those results. For all example there are notebooks showing how the E value distributions are calculated as well as kinetic plots:

For dynamic FRET:

https://github.com/duderstadt-lab/mars-tutorials/blob/master/Example_workflows/FRET/dynamic/dynamic_FRET_example.ipynb

For No aex dynamic FRET:

https://github.com/duderstadt-lab/mars-tutorials/blob/master/Example_workflows/FRET/no_acceptor_excitation/no_acceptor_excitation_FRET_example.ipynb

In the example pages on the Mars documentation website, we have provided some ideas for integration of HHM algorithms commonly used for smFRET analysis into Mars workflows using Python. We created an issue (https://github.com/duderstadt-lab/mars-tutorials/issues/1) to track this and hope users reach out and request this feature. Then we can explore it further. This is a very big topic and we feel it goes far beyond the scope of the current manuscript.

We have revised the smFRET workflow text of the manuscript accordingly to include a description of the newly collected dataset and treatment of dynamic smFRET. We have included new figures showing the results of analysis of the new datasets and included the experimental details in the methods scripts of the revised manuscript.

b. The analysis pipeline was designed to filter out traces with large intensity changes that are not due to photo-bleaching. Large intensity changes might be interesting if one of the FRET dyes experiences quantum-yield changes that are unrelated to FRET (e.g., smFRET involving Cy3 and Cy5). It could be interesting in the proper context, and hence should allow the user to change the data rejection criteria, in a context-dependent manner.

This is an interesting point. Two change point commands are included in Mars. One only looks for the single most likely change point positions in the trace, which is used to detect the bleach positions. The other, normal change point command, will detect all state transitions based on an acceptable false positive rate provided and level of expected noise. The normal change point command can be used on the whole trace or marked regions to perform exactly the type of analysis you have mentioned.

In the case of the new dynamic FRET example dataset, we found leveraging the prior knowledge that we expected two state behavior yielded more accurate dwell time distributions using a simple threshold-based script. We have designed the workflows and Mars so that change point can easily be included and used for other types of analysis.

The change point tables generated could be used to develop further specific rejection criteria. The code used for generation of the dwell time distributions provide a starting point. This could also be easily accomplished in a groovy script. We hope we understood the reviewers comment properly.

c. Correction factors 1: the text relies on the correction factors presented by the multi-lab work of Hellenkamp et al. (2017). That approach uses the Lee et al., (2005) approach for calculating the γ correction factor, which relies on the linear dependence of the inverse of the mean stoichiometry on the mean proximity ratio, of multiple single-population smFRET-ALEX measurements. While this correction factor procedure is used by many and is very common, it does suffer from assuming that both donor and acceptor fluorescence quantum yields change in the same manner when positioning the same donor and acceptor dyes at different positions along the same molecule (a dsDNA in this case). This might not be true, as different labeling positions introduce different dye microenvironments, which may introduce different quenching rates that are unrelated to FRET. In other words, it is possible (and not uncommon at all) that different molecules with different FRET values will also be associated with different γ factors. There are alternatives that can assist in calculating per-population γ factors, which require knowledge of the donor and acceptor fluorescence lifetimes per population. However, this is not yet possible in immobilized TIRF camera-based smFRET. Therefore, in page 10 of the bioRxiv preprint, when the authors present the γ correction factor, I suggest they add: "… , under the assumption that placing the same dyes at different bases along the DNA molecule does not influence the ratio of the acceptor and donor fluorescence quantum yields".

We thank the reviewer for pointing out this important assumption in our analysis. We have added the suggested text in the description for the corrections on page 10 as suggested.

d. Correction factors 2: the text relies on the correction factors presented by the multi-lab work of Hellenkamp et al., (2017). That approach uses the Lee et al., (2005) approach for calculating the γ correction factor. However, Hellenkamp et al., (2017) and not Lee et al., (2005) is cited. I ask the authors to cite Lee et al., (2005), when providing the explanation on the β- and γ-factor calculations.

This was an oversight on our part. We have added the Lee et al., (2005) reference in the revised manuscript. We thank the reviewer for pointing this out.

e. When explaining the FRET-related calculations, sometimes the variables are not explained: (i) in steps 4 and 5, explicitly explain the readers what are FA|A and FD|D; (ii) "FA|D stores the fully corrected acceptor fluorescence intensity value"; (iii) explain for FRET novices what do the variables in equations 1 and 2 mean.

Thank you for pointing out these errors. We have included more explanations about each of the variables introduced in the FRET calculations section of the revised manuscript.

f. At the bottom of page 6 of the bioRxiv preprint, the authors provide their estimations of the FRET efficiency values, and it seems that the accuracy is high, if comparing the values to the mean values reported by Hellenkamp et al. (2017). However, the precision of the reported values (E1-lo = 0.14 {plus minus} 0.13 and E1-mid = 0.51 {plus minus} 0.09) are low compared to the precisions in Hellenlamp et al. (E1-lo = 0.15 {plus minus} 0.02 and E1-mid = 0.56 {plus minus} 0.03). Why such imprecisions? Please explain the readers in the text.

First of all, we apologize for a mistake in the text where these values are reported that has added to this confusion. It is not the standard error of the mean we reported, but simply the standard error (standard deviation). We reported the standard deviation from a gaussian fit of the two E value distributions for 1-lo and 1-mid. This precision estimate reflects the broadness of the E value distributions themselves. The wide E value distributions we observe are similar to Hellenkamp et al. This can be seen clearly by comparing our E value distributions to the ones displayed in their figures. However, they do not report this standard deviation. Instead, Hellenkamp et al., report the standard deviation of the mean values of E reported by many labs. The individual values used for this calculation of precision each represent the best estimate or center position of the E value peaks obtained for a given lab. As a consequence, the standard deviation is small from these values.

In the process of revising the smFRET examples, we reanalyzed the static FRET dataset with the new workflow and obtained the updated values E1-lo = 0.16 ± 0.11 and E1-mid = 0.53 ± 0.13 (standard deviation). We have revised the manuscript to include these values because they reflect the results from the revised workflow and newly deposited example Molecule Archive in the mars-tutorials repository. We have also included the same values but with the estimate of the standard error of the mean: E1-lo = 0.16 ± 0.01 and E1-mid = 0.53 ± 0.01 for the ~250 molecules in each the dataset.

We have included an explanation of the differences in precision we observe compared to Hellenkamp et al., due to difference in the calculations performed. We thank the reviewer for pointing this out.

Reviewer #2 (Recommendations for the authors):Technical issues encountered:1. Overall the step-wise tutorial on the website is well organized, but can be further improved. Particularly, the expected results/output files/windows should also be included, similar as the parameter setting for the input part. Sometimes it's unclear what the correct output format should be from each operation.

We have revised the workflow to further clarify the input and output of each of the steps. We have created an overview figure and split the process into two phases. The first phase is focused on image processing to obtain three Molecule Archives (FRET, DO, and AO). The second phase performs a bunch of operations on the merged Molecule Archive with a streamlined set of numbered scripts (https://github.com/duderstadt-lab/mars-tutorials/tree/master/Example_workflows/FRET/scripts). We have included screenshots of the dialogs presented from the scripts and make it clearer what the expected outputs are. We hope the full example video on YouTube clarifies any further questions (https://www.youtube.com/watch?v=JsyznI8APlQ).

2. We noticed Mac and Windows systems could encounter different issues. Taking the FRET analysis as an example, we had to switch between the Windows and Mac systems to get it through completely, because certain steps worked in one but not the other.

We are very sad to hear you had to switch between computer systems. This problem never occurred during our tests. We apologize for the issues. We have worked to improve and address each step you have mentioned below.

One frequent issue people encounter is an out-of-date copy of Fiji with incorrect jars. Therefore, to avoid those issues, we would strongly suggest downloading and installing a new copy of Fiji and updating that with the Mars update site as described in the installation instructions (now in the manuscript as well as on the Mars documentation page). See out point-by-point comments for the other issues below.

For example:(1) When we tried to convert the example FRET video to contain channel information using script 1, the Windows system generated a blank image. However, it ran correctly on Mac.

This particular script was written using the ImageJ1 API. This could account for the problems you encountered. We rewrote the script using the ImageJ2 API and Imglib2 (https://github.com/duderstadt-lab/mars-tutorials/blob/master/Example_workflows/FRET/scripts/static_FRET_reformat_video.groovy).

This improved the performance of the script and we hope it is now more reliable. We have tested it on windows and mac without problems. Just in case, when you observe a blank image are you sure it didn’t work or is the contrast just not adjusted? It can happen that you see a black image due to the wrong contrast settings. This can be adjusted using Image > Adjust > Brightness and Contrast…

(2) There was some problem with transforming the ROIs to the right part of the split view on a Mac system. We used the same parameters as the instruction, but all the coordinates disappeared after doing Transform ROIs. But it worked well on a windows system.

We have done further testing with the Transform ROIs command in the process of revising the smFRET example. We have now also added an inverse transform option, so the coordinates don’t need to be changed during the workflow. We hope this addresses the problem. We haven’t see any differences in how the commands works when comparing windows and mac.

(3) On a Mac system, after running peak tracer or molecular integrator, we were not able to generate a GUI interface containing the actual output, but a console saying the analysis is successful.

This is very unfortunate and unexpected. This can occur if JavaFx or some of the Mars jars are not installed on the mac system. We have worked really hard with the Fiji development team and now JavaFx is included with Fiji for mac. We would suggest trying a new copy of Fiji, freshly downloaded from https://fiji.sc with Mars installed as described in the installation instructions. However, these problems should have generated error messages.

If the detection threshold is too high peaks might not be tracked for many time points and this could result in no molecule found that have tracks long enough. In this case, it will report that the tool rand successfully, but there will be not Molecule Archive created because not results were found. Could this have been the issue?

The Molecule Integrator requires a set of ROIs in the ROI Manager to work. Was there a set of ROIs added to the ROI manager using the Peak Finder before running the command? If not, the message in the console should report the problem that no ROIs were found.

The successful statement at the bottom of log after a command has finished just indicated it ended without obvious serious errors. It doesn’t guarantee and output if the settings don’t produce one.

3. It's very confusing that the output file of the converted FRET data is a composite image of two channels. It is not explained what each of the channels is until explaining the output file of the molecular integrator step. Echoing my point 1, it is necessary to explain to the users the format of the output files. The composite image makes it appear that two channels are the same without any explanation, and each channel is the sum of the red and green signal.

We now see our mistake in not explaining this clearly. We have revised the documentation to include a better description of the video input with the two channels in the OME section. We have also included a YouTube video for the dynamic FRET example that has a similar video input. In the video we explain the different channels and what information they contain. Thank you very much for pointing out his issue.